# Retrograde transport of CDMPR depends on several machineries as analyzed by sulfatable nanobodies

Dominik P Buser ⓘ, Gaétan Bader, Martin Spiess ⓘ

**Retrograde protein transport from the cell surface and endosomes to the TGN is essential for membrane homeostasis in general and for the recycling of mannose-6-phosphate receptors (MPRs) for sorting of lysosomal hydrolases in particular. We used a nanobody-based sulfation tool to more directly determine transport kinetics from the plasma membrane to the TGN for the cation-dependent MPR (CDMPR) with and without rapid or gradual inactivation of candidate machinery proteins. Although knockdown of retromer (Vps26), epsinR, or Rab9a reduced CDMPR arrival to the TGN, no effect was observed upon silencing of TIP47. Strikingly, when retrograde transport was analyzed by rapamycin-induced rapid depletion (knocksideways) or long-term depletion by knockdown of the clathrin adaptor AP-1 or of the GGA machinery, distinct phenotypes in sulfation kinetics were observed, suggesting a potential role of GGA adaptors in retrograde and anterograde transport. Our study illustrates the usefulness of derivatized, sulfation-competent nanobodies, reveals novel insights into CDMPR trafficking biology, and further outlines that the selection of machinery inactivation is critical for phenotype analysis.**

## Introduction

Retrograde transport of proteins from the cell surface and endosomes to the TGN is critical for membrane homeostasis and to retrieve components of anterograde transport machineries. Proteins recycled back to the TGN comprise transport receptors for lysosomal hydrolases, processing enzymes, SNAREs (soluble N-ethylmaleimide–sensitive fusion factor attachment receptors), nutrient transporters, and a subset of other intracellular transmembrane proteins with diverse functions (Bonifacino & Rojas, 2006; Johannes & Popoff, 2008). In addition, extracellular bacterial and plant toxins exploit the retrograde transport machineries of target cells. One of the most thoroughly studied cargoes retrieved from endosomes to the TGN are the cation-dependent and cation-independent mannose-6-phosphate receptors (CDMPR and CIMPR), involved in efficient anterograde transport of lysosomal acid-

hydrolases from the TGN to endosomes (Ghosh et al, 2003a). After cargo unloading in the mildly acidic endosomal environment, MPRs are recycled to the TGN apparently by several retrograde transport machineries from both early/recycling and late endosomes (Sandvig & van Deurs, 2002; Bonifacino & Rojas, 2006; Pfeffer, 2009; McKenzie et al, 2012). The estimated copy number of CDMPR and CIMPR in HeLa cells is ~660,000 and ~310,000 per cell, respectively (Itzhak et al, 2016). Recently, cell surface MPR has been recognized as an efficient and potential therapeutic platform for targeted degradation of extracellular and transmembrane proteins using hexasaccharide–anti-target conjugates by shuttling the target to lysosomes, whereas MPR is retrieved to the TGN or plasma membrane (Banik et al, 2020). Understanding transport routes and sorting machineries involved in MPR trafficking is central to modulate this pathway.

The most prominent sorting machinery reported to mediate transport of MPRs to the TGN is the retromer complex, a pentameric protein assembly comprising Vps (vacuolar protein sorting) and SNX-BAR (sorting nexin-Bin/Amphiphyisin/Rvs) subunits (Bonifacino & Hurley, 2008; Seaman, 2012; Gallon & Cullen, 2015). The core complex, termed retromer, consists of the heterotrimer Vps26-Vps29-Vps35 that transiently associates with the tubulation subcomplexes composed of SNX1 or SNX2, and SNX5 or SNX6. Other SNX-BAR proteins, including SNX3, have been shown to mediate endosome-to-TGN transport of Wntless (WLS) (Belenkaya et al, 2008; Harterink et al, 2011). The precise sites from which retromer complexes operate remain to be defined: Vps35 of retromer was shown to be recruited by Rab7a, a marker of late endosomes, but SNX-BAR components of the multimeric complex bind via their Phox homology (PX) domain to phosphatidylinositol 3-phosphate (PI3P), a phospholipid enriched on early endosomes (Cozier et al, 2002; Carlton et al, 2004, 2005; Rojas et al, 2008; Seaman et al, 2009). It was thus proposed that MPR sorting by retromer complex is a progressive process coupled to endosomal maturation during the Rab5-to-Rab7 switch (Rojas et al, 2008). Some uncertainty also exists about cargo recognition by Vps and/or SNX-BAR subunits. Previously, it was reported that Vps subunits serve as cargo adaptors for the cytoplasmic domain of CIMPR (Nothwehr et al, 2000; Seaman, 2007; Fjorback et al, 2012; Lucas et al, 2016; Cui et al, 2019; Suzuki et al, 2019). Two recent independent studies rather suggest that SNX1/2 and SNX5/6 mediate cargo recognition and

Biozentrum, University of Basel, Basel, Switzerland

Correspondence: dominik-pascal.buser@unibas.ch

retrieval of CIMPR (Kvainickas et al, 2017; Simonetti et al, 2017). Their results not only showed that SNX-BAR dimers associate with a WLM motif in the cytoplasmic tail of CIMPR, but also that Vps35 depletion, unlike depletion of SNX-BARs, did not cause receptor misdistribution from juxtanuclear to peripheral compartments. This observation is in disagreement with previous results showing a prominent mis-localization of CIMPR to endosomes upon silencing of Vps26 or Vps35 (Arighi et al, 2004; Seaman, 2004).

A further pathway for retrograde transport of MPRs was de-scribed to involve Rab9a and the adaptor TIP47 (tail-interacting protein of 47 kD). Using a cell-free system, it was shown that Rab9a recruits TIP47 to late endosomes and that interference with GTPase-effector function resulted in severe impairment of trans-port of MPRs (Lombardi et al, 1993; Diaz & Pfeffer, 1998). However, TIP47 was since identified to be a component of lipid droplets involved in their biogenesis (Bulankina et al, 2009) and an addi-tional role in retrograde transport was not independently repro-duced (Medigeshi & Schu, 2003).

Another mechanism for MPR retrieval to the TGN involves clathrin-coated vesicles (CCVs) with the adaptor protein (AP)–1 complex and/or epsinR. AP-1 has a generally accepted role in anterograde transport of MPRs from the TGN to endosomes in cooperation with GGA (Golgi-localized, γ-adaptin ear-containing, Arf-binding) proteins (Doray et al, 2002; Ghosh & Kornfeld, 2004; Sanger et al, 2019). Unlike retromer complexes, AP-1 has a dual distribution both at the TGN and on early endosomes (Le Borgne et al, 1996; Seaman et al, 1996; Meyer et al, 2000). Inactivation of AP-1 was found to result in a dispersed MPR localization pattern towards the periphery of cells (Meyer et al, 2000; Robinson et al, 2010), similar to the phenotype of retromer complex inactivation, sug-gesting a role in retrograde transport. EpsinR, an interactor of AP-1, was also shown to have a role in CIMPR retrieval to the TGN (Hirst et al, 2004; Saint-Pol et al, 2004), although receptor distribution did not significantly change in epsinR-depleted cells. It is puzzling, why epsinR and AP-1 have different effects on MPR localization when depleted, whereas they seem to depend on each other for incor-poration into CCVs, depletion of one reducing the CCV content of the other (Hirst et al, 2004, 2015).

GGAs also localize to both TGN and endosomes (Boman et al, 2000; Ghosh et al, 2003b). Yet, they have been implicated mainly in anterograde transport. Rapid depletion of GGA2 by knocksideways specifically depleted lysosomal hydrolases and their receptors (MPRs and sortillin) from CCV contents, whereas knocksideways of AP-1 affected also a number of SNAREs and additional membrane proteins (Hirst et al, 2012). Depletion of lysosomal hydrolases from CCVs was more efficient upon inactivation of GGA2 and depletion of their receptors more efficient upon inactivation of AP-1. This result suggested a role of GGAs primarily in anterograde transport and of AP-1 in both directions.

To analyze retrograde transport machinery, most studies used fluorescence microscopy to monitor changes in MPR localization relative to TGN-resident or endosomal markers by statistical steady-state image analysis. A few laboratories imaged antibody uptake to follow retrograde transport from the cell surface to the TGN (e.g., Breusegem and Seaman [2014]), whereas Johannes and colleagues used sulfation as a specific modification of the trans-Golgi/TGN to probe Golgi arrival of Shiga toxin B-chain (STxB)

tagged with a sulfation motif or of antibodies derivatized with sulfatable peptides (Saint-Pol et al, 2004; Popoff et al, 2009). However, the disadvantage of conventional divalent antibodies is that they are rather large and can crosslink their antigens and thus potentially alter their trafficking. In contrast, monomeric protein binders, such as nanobodies, are monovalent and small. Nano-bodies are easily derivatized with sequence tags and fluorescent or enzymatic protein domains and can be produced in bacteria.

To study retrograde traffic, we have previously established a versatile toolkit of functionalized anti-GFP nanobodies (Buser et al, 2018; Buser & Spiess, 2019). In particular, we generated nanobodies containing tyrosine sulfation (TS) sites to monitor their arrival in the trans-Golgi/TGN. Using cell lines stably expressing EGFP-CDMPR or EGFP-CIMPR, we determined the transport kinetics of these re-ceptors from the cell surface to the TGN. In addition, we used the knocksideways system developed by Robinson et al (2010) to an-alyze the contribution of AP-1 upon rapid depletion. The system is based on rapamycin-induced heterodimerization between the γ-subunit of AP-1 fused to FKBP12 (FK506-binding protein of 12 kD) and the FKBP–rapamycin-binding domain (FRB) of mammalian target of rapamycin anchored in the outer mitochondrial mem-brane as a trap (Mitotrap). Upon rapid inactivation of AP-1, a robust reduction of sulfation kinetics by approximately on third was ob-served, confirming a significant contribution of AP-1/CCVs in ret-rograde transport of MPRs (Buser et al, 2018).

In the present study, we applied this tool of sulfatable nano-bodies to analyze plasma membrane-to-TGN transport kinetics of CDMPR to define the contribution of individual different sorting machineries in parallel on retrograde transport in living cells. We could confirm retrograde transport activity of retromer and epsinR as well as a considerable involvement of Rab9a, but not of TIP47. Unexpectedly, silencing of GGA1-3 using RNAi also reduced CDMPR arrival at the TGN, suggesting a role of GGAs in endosome-to-TGN rather than anterograde transport. Conversely, acute inactivation of GGA2 by knocksideways did not affect CDMPR transport from the cell surface to the compartment of sulfation, suggesting possible indirect effects by RNAi.

# Results

## Functionalized nanobodies to analyze retrograde transport of CDMPR in cells depleted of candidate machineries

To study retrograde traffic to intracellular compartments including the TGN, we have previously established a versatile toolkit of functionalized anti-GFP nanobodies (Buser et al, 2018; Buser & Spiess, 2019). They can be used to label GFP-tagged proteins of interest at the cell surface and follow their route to endosomes, the TGN, and back to the plasma membrane. Here, we used anti-GFP nanobodies (VHH, variable heavy-chain domain of heavy-chain–only antibody) modified with a hexahistidine tag for purification, a T7 and an HA tag for immunodetection, a biotin acceptor peptide for biotinylation, and sequences conferring tyrosine sulfation (VHH-2xTS) or red fluorescence (VHH-mCherry) (Fig 1A). These func-tionalized nanobodies were bacterially expressed and isolated to

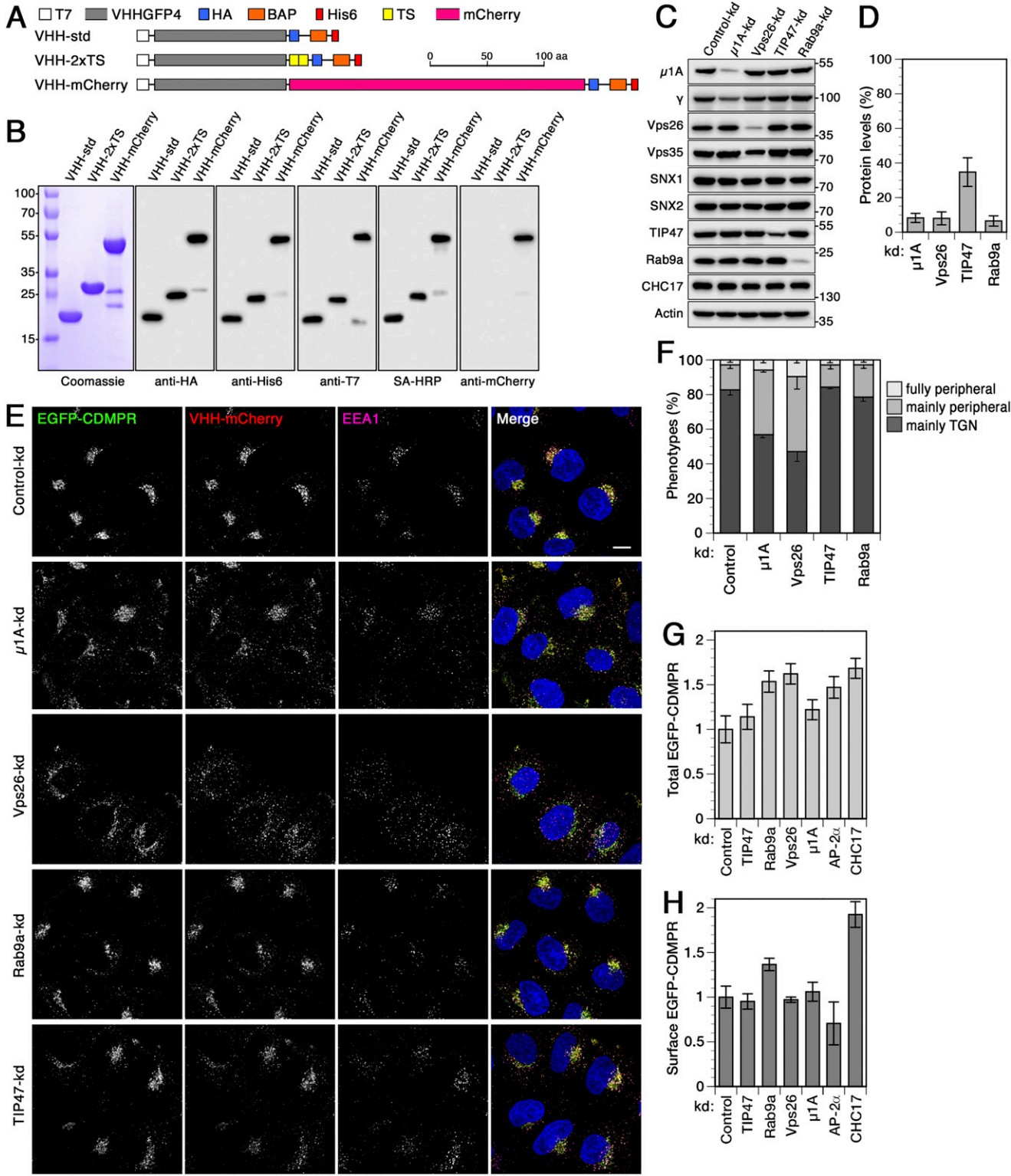

**Figure 1. Functionalized nanobodies to analyze retrograde transport of CDMPR upon silencing of adaptor protein-1, retromer, Rab9a, or TIP47.**
**(A)** Schematic representation of the functionalized nanobodies. The standard nanobody (VHH-std) consists of the GFP-specific VHH domain, T7 and HA epitope tags, a biotin acceptor peptide (BAP), and a hexahistidine (His6) purification tag. Other nanobodies in addition contain two tyrosine sulfation sequences (VHH-2xTS) or mCherry (VHH-mCherry). Scale bar in aa. **(B)** Bacterially expressed and purified nanobodies (30 μg) were analyzed by SDS-gel electrophoresis and Coomassie staining (left). Immunoblot analysis of nanobodies (10 ng) with antibodies against the HA, His6, T7, or mCherry epitopes, or with streptavidin-HRP (SA-HRP). Marker proteins with molecular weights in kilodalton are shown on the left. As previously reported (Buser et al, 2018; Buser & Spiess, 2019), mCherry-containing nanobodies are slightly susceptible to clipping between the VHH and mCherry domains. **(C)** HeLa cells were transfected with non-targeting siRNA or siRNAs targeting μ1A-adaptin, Vps26, TIP47, or

high purity, and shown to be efficiently immunodetected using epitope tag antibodies or streptavidin-HRP (Fig 1B).

A well-established phenotype of retrograde transport deficiency on MPR traffic is the redistribution of the receptor from juxtanuclear to more peripheral compartments. To analyze the contribution of individual MPR retrieval routes to the TGN, we depleted machinery components of the AP-1/clathrin-, the retromer-, and the Rab9a/TIP47-dependent pathways by RNA interference using well-established siRNAs. Inactivation of the AP-1 complex was achieved by depleting the μ1A-subunit of the heterotetrameric adaptor complex (Hirst et al, 2003, 2005, 2009). The retromer complex was inactivated by silencing the subunit Vps26 of the cargo–selective complex (Popoff et al, 2007, 2009). Rab9a and TIP47, which do not form a stable complex, were knocked down separately (Ganley et al, 2004; Reddy et al, 2006; Bulankina et al, 2009; Kucera et al, 2016). All these proteins could be robustly depleted by >85% (Fig 1C and D), except TIP47 which was consistently reduced by ~65%, similar to the depletion efficiencies for TIP47 in the literature where they had been reported to produce clear effects on MPR traffic in vitro and in vivo (Diaz & Pfeffer, 1998; Ganley et al, 2004). As previously observed, depletion of μ1A or Vps26 caused a concomitant reduction of its complex partners γ-adaptin or Vps35, respectively (Fig 1C) (Meyer et al, 2000; Arighi et al, 2004), whereas the retromer-associated SNX-BAR proteins SNX1 and SNX2 were not affected upon depletion of Vps26 (Arighi et al, 2004; Rojas et al, 2007). Depletion of components of one pathway did not affect expression levels of proteins associated with other retrograde transport routes to the TGN (Fig 1C).

We silenced the above machinery components in HeLa cells stably expressing EGFP-CDMPR and analyzed its steady-state localization by fluorescence microscopy (Fig 1E) to test for mislocalization to endosomal compartments, a well-documented phenotype thought to result from defective endosome-to-TGN retrieval (Meyer et al, 2000; Arighi et al, 2004; Seaman, 2004; Popoff et al, 2009; Robinson et al, 2010; Hirst et al, 2012). In addition, we added VHH-mCherry nanobodies to the cells for 1 h before fixation to specifically detect the mature pool of the receptor cycling between surface, endosomes, and TGN (similar to previous antibody uptake experiments [Meyer et al, 2000; Robinson et al, 2010]). To measure the extent of mislocalization of EGFP-CDMPR to peripheral compartments, we used a semi-quantitative approach classifying the CDMPR staining patterns of individual cells as "mainly TGN," "mainly peripheral," and "fully peripheral" (Fig 1F) as previously applied by Cullen and colleagues (Wassmer et al, 2007; Simonetti et al, 2017). Knockdown of AP-1 or retromer caused a pronounced shift of both receptor and imported nanobody from

the TGN to peripheral punctae as compared with control cells. We observed an approximately threefold increase in peripherally dispersed MPR-nanobody distributions in AP-1- and Vps26-depleted cells (Fig 1E and F), in agreement with previous analyses (e.g., Wassmer et al [2007]). In contrast, MPR localization was not significantly affected upon depletion of TIP47 or Rab9a.

Because CDMPR abundance, particularly at the plasma membrane, affects nanobody uptake and sulfation, we determined total and surface levels in silenced cells biochemically or by flow cytometry. Knocking down any of these four components did not cause any apparent change in plasma membrane levels of EGFP-CDMPR as assessed by nanobody (VHH-2xTS) binding only to the surface receptors at 4°C followed by immunoblotting (Fig S1). As a positive control, depletion of clathrin heavy chain (CHC17), which is required for clathrin-mediated endocytosis, produced the expected clear increase in surface EGFP-CDMPR (Fig S1). Interestingly, knockdown of the AP complex 2 (AP-2) did not mimic the phenotype of CHC17 depletion, in line with the observation that an AP-2 knockdown also did not cause cell surface accumulation of CIMPR (Dugast et al, 2005; Keyel et al, 2008; Tobys et al, 2021). To more quantitatively determine total and surface levels of EGFP-CDMPR in machinery-depleted cells, we performed flow cytometry to quantify the mean fluorescence intensity of GFP and of surface-bound VHH-mCherry, respectively (Fig 1G and H). RNAi-mediated depletion of Rab9a and Vps26 caused a significant increase of total EGFP-CDMPR levels by ~60%, probably by reducing lysosomal receptor degradation (Fig 1G). Surface receptor abundance was only augmented in Rab9a-silenced cells by ~40% (Fig 1H). Silencing of TIP47 or μ1A did not affect CDMPR levels.

To make sure that depletion of potential machinery components do not generally affect sulfation efficiency, on which our assay critically depends, HeLa cells stably expressing a sulfatable form of the secretory protein α1-protease inhibitor (A1Pi) were labeled with [35S]sulfate. No change in sulfation of A1Pi was observed for any of these protein knockdowns (Fig S2A and B).

### Retrograde transport of CDMPR to the TGN is affected by depletion of Rab9a or Vps26, but not of TIP47

To more directly, more sensitively, and more quantitatively assay endosome-to-TGN transport in control and knockdown cells, we used VHH-2xTS nanobodies containing sites for tyrosine sulfation. This allows us to correlate appearance of nanobody sulfation with TGN arrival and residence time in this compartment. Cells stably expressing EGFP-CDMPR were silenced for one of the candidate retrograde machinery components or, as a control, transfected with

---

Rab9a. 3 d after transfection, the cells were subjected to immunoblot analysis with antibodies against the indicated proteins. **(D)** To determine the knockdown (kd) efficiency, the residual protein was quantified in percent of the value after control-kd (mean and SD of three independent experiments). **(E)** HeLa cells stably expressing EGFP-CDMPR were depleted of μ1A-adaptin, Vps26, TIP47, or Rab9a as in (C). Cells were incubated for 1 h at 37°C with full medium containing 5 μg/ml VHH-mCherry (~0.1 μM), fixed, stained for EEA1 and nuclei (DAPI, blue), and imaged by fluorescence microscopy. Bar: 10 μm. **(F)** Quantitation of the percentage of cells displaying the CDMPR localization phenotypes "mainly TGN," "mainly peripheral," or "fully peripheral" as in Wassmer et al (2007) and Simonetti et al (2017). For each condition, random frames with a total of 136–140 cells were scored from three independent experiments. **(G, H)** Total and surface EGFP-CDMPR levels in RNAi-silenced cells were quantified by flow cytometry. Cells were incubated for 30 min at 4°C with VHH-mCherry for exclusive binding to EGFP-CDMPR at the cell surface, washed, dissociated and analyzed for GFP and mCherry fluorescence to determine the levels of total (G) and surface EGFP-CDMPR (H), respectively. Median fluorescence intensities above background of parental HeLa cells without EGFP-CDMPR of each condition were normalized to the average of cells treated with non-targeting control siRNA. For each condition, 50,000 cells were analyzed in each experiment (mean and SD of three independent experiments).

non-targeting siRNA. The cells were then incubated with media containing VHH-2xTS for up to 75 min while labeling with [$^{35}$S]sulfate. In control cells, nanobody binding to EGFP-CDMPR and uptake reached its maximum within little more than 30 min and 50% after about 15 min (Fig 2A and B, open squares). Sulfation started only after a lag time of ~15 min and had not yet reached saturation after 75 min (Fig 2B, filled squares), in full agreement with our previous report (Buser et al, 2018). The difference between uptake and sulfation curves reflects the time of transport to the TGN.

The most thoroughly analyzed sorting machinery in endosome-to-TGN retrieval of MPRs is the canonical retromer complex (Arighi et al, 2004; Seaman, 2004; Wassmer et al, 2007; Burd & Cullen, 2014). Knocking down Vps26 as a core component of retromer caused a significant reduction of the rate of sulfation, indicating only ~50% of nanobody transport to the TGN after 75 min, whereas uptake was not considerably affected (Fig 2A and B). These results support a contribution of Vps26 in retrograde transport of CDMPR, confirming previous observations by different laboratories for CIMPR using immunofluorescence and antibody uptake assays (Arighi et al, 2004; Seaman, 2004, 2007; Popoff et al, 2007, 2009; McKenzie et al, 2012). Notably, we obtained a similar extent of transport impairment for EGFP-CDMPR as previously reported for STxB using single-time point sulfation experiments (Popoff et al, 2007, 2009).

Upon depletion of Rab9a, kinetics of nanobody uptake remained unaffected, whereas sulfation was modestly, but significantly reduced (Fig 2C and D). This effect is consistent with previous observations based on image analysis and antibody uptake for chimeric CIMPR and furin (Seaman et al, 2009; Chia et al, 2011). In these previous reports, transport to the TGN was reduced by up to 50% upon Rab9 knockdown, whereas we could only observe a reduction in signal of ~25% after 75 min, a difference that might be due to the method used. Taking the increased total and surface levels of EGFP-CDMPR in silenced cells into account (Fig 1H and G), the observed reduction of ~25% after 75 min might be an underestimation of the effect of Rab9a on retrograde CDMPR transport.

Whereas depletion of Rab9a or Vps26 both significantly affected transport of CDMPR from the plasma membrane to the compartment of sulfation, lysosomal hydrolase delivery—as measured by increased secretion of recombinant myc-tagged procathepsin D into the medium—was only affected by knockdown of Vps26, but not of Rab9 (Fig 2E and F). This is consistent with previous studies showing that CRISPR/Cas9–mediated deletion of Rab9a had no effect on cathepsin delivery to lysosomes, whereas lysosomal delivery of hydrolases was defective in retromer- or Vps26-deficient cells (Bugarcic et al, 2011; Fuse et al, 2015; Cui et al, 2019; Homma et al, 2019). As additional control, we analyzed secretion of transfected His6/myc-tagged PAUF (pancreatic adenocarcinoma up-regulated factor), a cargo of CARTS (carriers of the TGN to the cell surface) that bypass membranes related to Vps26 or Rab9a function (Wakana et al, 2012). Indeed, PAUF secretion was not altered by either knockdown (Fig 2G and H), confirming that the effect seen on procathepsin D is specific for Vps26.

Because TIP47 was proposed to mediate Rab9a-dependent MPR transport, knockdown should produce a similar effect as Rab9a depletion. However, kinetics of nanobody uptake and sulfation by EGFP-CDMPR remained unchanged in TIP47 knockdown cells (Fig 2I and J). Our sulfation experiments add an additional method to those used previously to evaluate retrograde transport kinetics of CDMPR with and without TIP47, again with a negative result.

## AP-1 contributes to both retrograde and anterograde transport of CDMPR

To investigate the contribution of AP-1 in this process, we have previously analyzed the role of AP-1 in CD- and CIMPR transport to the TGN by rapid inactivation of AP-1 using knocksideways (Buser et al, 2018). Rapid depletion showed a reduction of approximately one third in the rate of sulfation, demonstrating a significant contribution of AP-1/clathrin in endosome-to-TGN transport of the MPRs. To test the outcome with AP-1 silencing in the same manner as applied to analyze the contribution of the other potential machineries above, we also performed the transport assay upon siRNA knockdown of μ1A.

Surprisingly, we did not observe a reduction of the kinetics and the extent of nanobody sulfation as expected from the more peripheral steady-state distribution of CDMPR in long-term AP-1–depleted cells. Instead, we found, after a similar lag phase as in all previous conditions, an ~2.5-fold increase in rate and extent of sulfation (Fig 3A and B). If this reflected transport directly, it would indicate increased retrograde transport activity by other mechanisms that even strongly overcompensated the loss of AP-1/clathrin–mediated transport. Yet, no increase in the levels of other machineries was detectable in μ1A knockdown cells (Figs 1C and S1) and no increase in general tyrosine sulfation (Fig S2A and B). The strong sulfation signal is also not the result of increased uptake of VHH-2xTS because the signals of cell-associated nanobody after loading at 37°C to steady-state or after binding only to cell-surface EGFP-CDMPR at 4°C were not increased in AP-1 knockdown compared with control cells (Figs 1H, 3A and B, and S1).

To rule out artefacts such as off-target effects of siRNA-mediated silencing of μ1A, we generated HeLa AP-1 knockout (ko) cells in which the γ1-adaptin genes were inactivated using CRISPR/Cas9. As expected, knockout cells displayed a complete loss of γ-adaptin staining in immunoblot and immunofluorescence analysis and a concomitant reduction of μ1A and σ1A subunits (Fig 3C and D). HeLa AP-1γ-ko cells recapitulated the phenotypes of μ1A knockdown cells. Whereas Golgi morphology remained unchanged, internalized anti-CIMPR antibody (as originally shown by Robinson et al [2010]) showed mainly peripheral localization that was largely rescued to a predominantly perinuclear Golgi localization upon re-expression of a γ-adaptin fusion protein (Fig 3E). Just like in the μ1A knockdown cells, sulfation of nanobody internalized by transfected EGFP-CDMPR was at least twofold higher after 75 min of labeling than in wild-type HeLa cells (Fig 3F and G).

However, we have previously observed that sulfation is not simply a detector of arrival in the compartment. Nanobody sulfation appears not to be sufficiently efficient to immediately and completely modify the sulfation sites as they enter the sulfation compartment. This explains why sulfation per nanobody depended on the target receptor: nanobodies taken up by EGFP-TGN46 showed considerably higher specific sulfation within 1 h than those captured by the EGFP-MPRs, even though maximal nanobody uptake was reached much later (Buser et al, 2018). This most likely reflects the residence time of these proteins in the sulfation

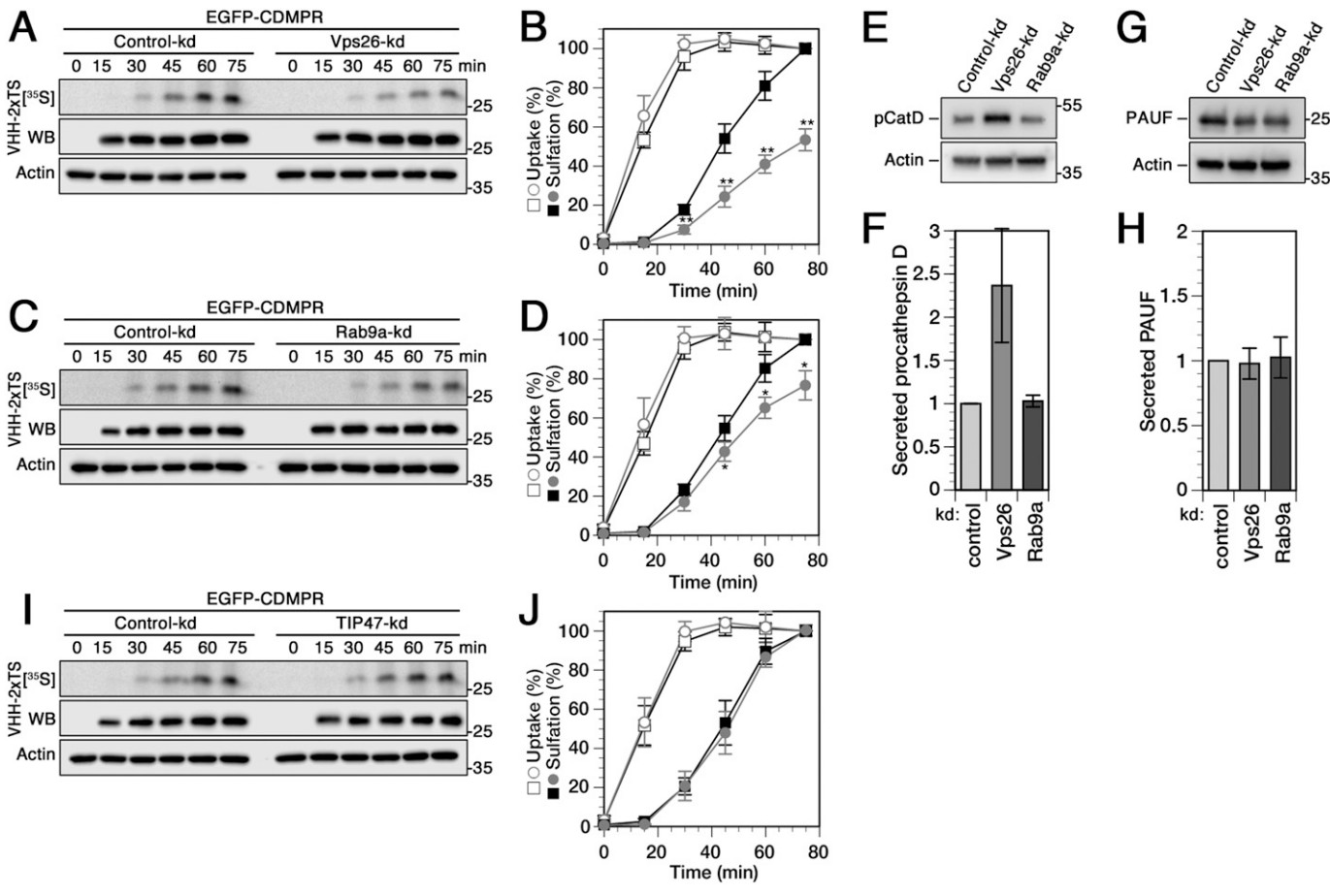

**Figure 2. Changes in retrograde transport kinetics of CDMPR to the TGN upon silencing Vps26, Rab9a, or TIP47.**
**(A, B, C, D)** Cells stably expressing EGFP-CDMPR were transfected with non-targeting siRNA (control-kd) or with siRNA silencing expression of Vps26 (A) or Rab9a (C) as described in Fig 1. The cells were labeled with [35S]sulfate for up to 75 min in the presence of 2 µg/ml VHH-2xTS. The nanobodies were isolated by Ni/NTA beads and subjected to SDS-gel electrophoresis followed by Western-blot (WB) analysis (anti-His6) and autoradiography ([35S]). In parallel, aliquots of the cell lysates were immunoblotted for actin as a control for the amount of cells used. Experiments as shown in panels (A) and (C) were quantified in panels (B) and (D), respectively, and presented as the percentage of the value of control-kd cells after 75 min (mean and SD of three independent experiments; two-sided t test: *P < 0.05; **P < 0.01). Control-kd is shown as black squares and target-kd as gray circles; uptake as open symbols, sulfation as filled symbols. **(E, F)** HeLa cells stably expressing His6/myc-tagged cathepsin D or PAUF were transfected with non-targeting control siRNA or RNAs silencing expression of Vps26 or Rab9a. Cells were incubated in serum-free medium supplemented with 5 mM mannose-6-phosphate for 2 h. Secreted procathepsin D (pCatD) or PAUF were collected by Ni/NTA beads and analyzed by immunoblotting with anti-myc antibodies. **(G, H)** Missorted procathepsin D or PAUF was quantified from immunoblots as shown in panels C and E, respectively, and normalized to the values of control knockdown cells (mean and SD of four [pCatD] and three [PAUF] independent experiments). **(I, J)** Cells stably expressing EGFP-CDMPR were transfected with non-targeting siRNA (control-kd) or with siRNA silencing expression of TIP47 and assayed and quantified as described above in panels (A, B, C, D) (mean and SD of three independent experiments).

compartment during which sulfate was continually incorporated into the nanobodies. A potential explanation of the observed hypersulfation is thus an increased residence time of the nanobody–EGFP-CDMPR complexes that still reached the TGN in the absence of AP-1. This is not unlikely because AP-1/clathrin is not only involved in retrograde retrieval to the TGN, but also in anterograde transport of MPRs out of the TGN to endosomes (Doray et al, 2002; Waguri et al, 2003; Ghosh et al, 2003a).

### Rapid AP-1 inactivation by knocksideways inhibits TGN exit of CDMPR

The bidirectional function of AP-1/clathrin in MPR traffic thus makes it impossible to directly compare nanobody sulfation

kinetics with other knockdown situations. To more directly demonstrate the effect of an anterograde transport block at the TGN on nanobody sulfation, we employed the AP-1 knocksideways cells (HeLa-AP1ks) established previously (Buser et al, 2018). AP-1 rerouting to mitochondria by rapamycin for 1 h shifted the steady-state distribution of EGFP-CDMPR to peripheral compartments as expected (Fig 4A and B). To demonstrate AP-1 dependence of TGN exit, we first loaded HeLa-AP1ks cells expressing EGFP-CDMPR with VHH-2xTS nanobody to steady-state during sulfate starvation, followed by [35S]sulfate labeling for up to 75 min. Upon addition of [35S]sulfate, there is a delay of 2–3 min for uptake and formation of 3'-phosphoadenosine-5'-phosphosulfate (PAPS) before incorporation of radioactivity gradually starts. To avoid these starting effects, rapamycin was added only 15 min after addition of

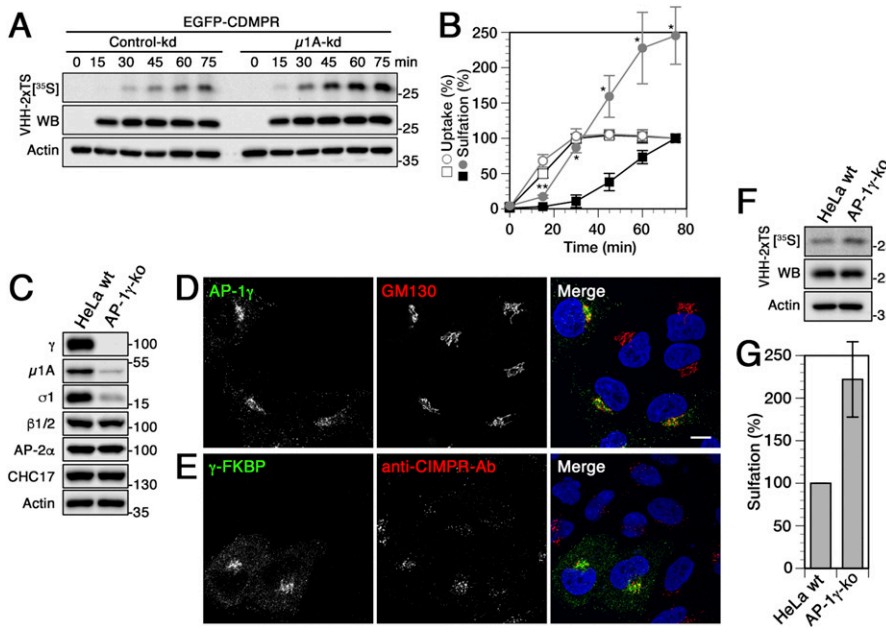

**Figure 3. Nanobodies transported to the TGN by CDMPR upon knockdown or knockout of adaptor protein (AP)-1 is hypersulfated.**
**(A, B)** Cells stably expressing EGFP-CDMPR were transfected with non-targeting siRNA (control-kd) or with siRNA silencing expression of µ1A as described in Fig 1. The cells were labeled with [$^{35}$S]sulfate for up to 75 min in the presence of 2 µg/ml VHH-2xTS and the nanobodies were isolated, analyzed, and quantified as in Fig 2 (mean and SD of three independent experiments; two-sided $t$ test: *$P < 0.05$; **$P < 0.01$). Control-kd is shown as black squares and µ1A-kd as gray circles; uptake as open symbols, sulfation as filled symbols. **(C)** Immunoblot analysis of parental HeLa cells (HeLa wt) and of a pool of γ-adaptin-knockout cells (AP-1γ-ko) generated with CRISPR/Cas9. Equal amounts of cell lysates were probed with antibodies against specific AP-1 subunits (γ, µ1A, σ1), β-adaptins of AP-1 and AP-2 (β1/2), AP-2α, clathrin heavy-chain (CHC17), and actin. **(D)** Parental HeLa cells and AP-1γ knockout cells were mixed and stained with antibodies targeting AP-1γ or GM130. γ-Adaptin staining was completely absent in knockout cells, whereas Golgi morphology remained intact. **(E)** AP-1γ knockout cells were transiently transfected with γ-FKBP (a fusion protein of γ-adaptin with FK506 binding protein; see the Materials and Methods section) and incubated with anti-CIMPR antibody for 1 h at 37°C. Cells were fixed and prepared for immunofluorescence microscopy by staining the recombinant γ-subunit and the internalized antibody. Non-transfected cells mostly displayed peripheral accumulation of anti-CIMPR antibody, whereas expression of γ-FKBP largely rescued perinuclear anti-CIMPR antibody localization. Nuclei were stained with DAPI (blue). Bar: 10 µm. **(F, G)** Parental HeLa and AP-1γ knockout cells were transiently transfected with EGFP-CDMPR, followed by [$^{35}$S]sulfate labeling for 75 min in the presence of 2 µg/ml VHH-2xTS. The nanobodies were isolated, analyzed, and quantified as in Fig 2 (mean and SD of three independent experiments).

[$^{35}$S]sulfate (illustrated in Fig 4C). Inactivation of AP-1 caused a more than twofold increase in sulfation rate compared with cells treated with vehicle only, to reach saturation within the next 15 min, much earlier than in control cells (Fig 4D and E). Whereas entry of CDMPR into the TGN is reduced by the rapid depletion of available AP-1 as we previously observed in retrograde transport experiment with the same cell line (Buser et al, 2018), the observed increase in sulfation in the present experiment thus reflects the accumulation of nanobody–EGFP-CDMPR in the compartment of sulfation because of reduced TGN exit. This offers itself also as an explanation of the hypersulfation in the TGN arrival assay of Fig 3.

To exclude the possibility that increased sulfation as observed by an AP-1 knockdown or knockout may be the consequence of mislocalization of sulfation machinery thereby causing sulfate incorporation to occur as early as into post-Golgi compartments, we tested whether the tyrosylprotein sulfotransferases (TPST1 and 2) were misdistributed to early endosomes. Neither TPST1- nor TPST2-EGFP were noticeably altered in localization to the Golgi in HeLa cells upon AP-1 knockdown or knockout (Fig S3A–D). No peripheral signal was detected, indicating that hypersulfation was not due to redistribution of sulfotransferases. As previously observed by Schu and colleagues (Meyer et al, 2000), AP-1 depletion slightly altered the originally even distribution pattern of EEA1-positive endosomes towards the perinuclear area.

## EpsinR and GGAs depletion both affect retrograde transport to the TGN

The exceptional role of AP-1 in mediating both anterograde and retrograde transport prompted us to consider how other APs that at least partially co-operate with AP-1 influence retrograde traffic of CDMPR, in particular epsinR and the GGA adaptors (GGA1–3). In

epsinR-depleted cells, isolated CCVs displayed a ~50% loss of incorporated AP-1, suggesting that AP-1 is to some extent dependent on epsinR for its incorporation into CCVs (Hirst et al, 2003, 2004). GGAs have been described to play a role in the packaging of MPRs into anterograde AP-1/clathrin carriers at the TGN (Doray et al, 2002; Ghosh & Kornfeld, 2004). This was further supported by the preferential depletion of lysosomal hydrolases and their receptors from CCVs upon GGA2 knocksideways (Hirst et al, 2012).

Levels of GGAs, individually or in combination, and of epsinR could be efficiently reduced by >85% by RNAi, whereas other sorting machineries remained unperturbed (Fig 5A and B). Also the steady-state levels of EGFP-CDMPR remained unaffected (Fig S1). Depleting epsinR in cells stably expressing EGFP-CDMPR did not considerably affect reporter-nanobody localization, with only a slight increase of MPRs redistributed to the periphery (Fig 5C and D). This observation is in agreement with previous findings showing no effect on the steady-state localization of CIMPR and furin upon silencing of epsinR (Hirst et al, 2004). Reducing GGA levels impacted the localization of MPR–nanobody similarly to the depletion of AP-1 (Fig 5C and D, compare with Fig 1E and F), in agreement with previous findings (Ghosh et al, 2003b). This phenotype is intriguing because it is rather characteristic for proteins mediating retrograde transport. Total and surface levels of CDMPR remained unaltered in cells depleted of epsinR and GGA adaptors as assessed by flow cytometry (Fig 5E).

Using our nanobody sulfation assay to determine the contributions of epsinR and GGAs on CDMPR transport, one would expect to find a reduction in sulfation kinetics to indicate inhibition of retrograde transport or hypersulfation to indicate inhibition of anterograde TGN exit. EpsinR-depleted cells showed no difference in nanobody uptake, but a strong impairment in retrograde

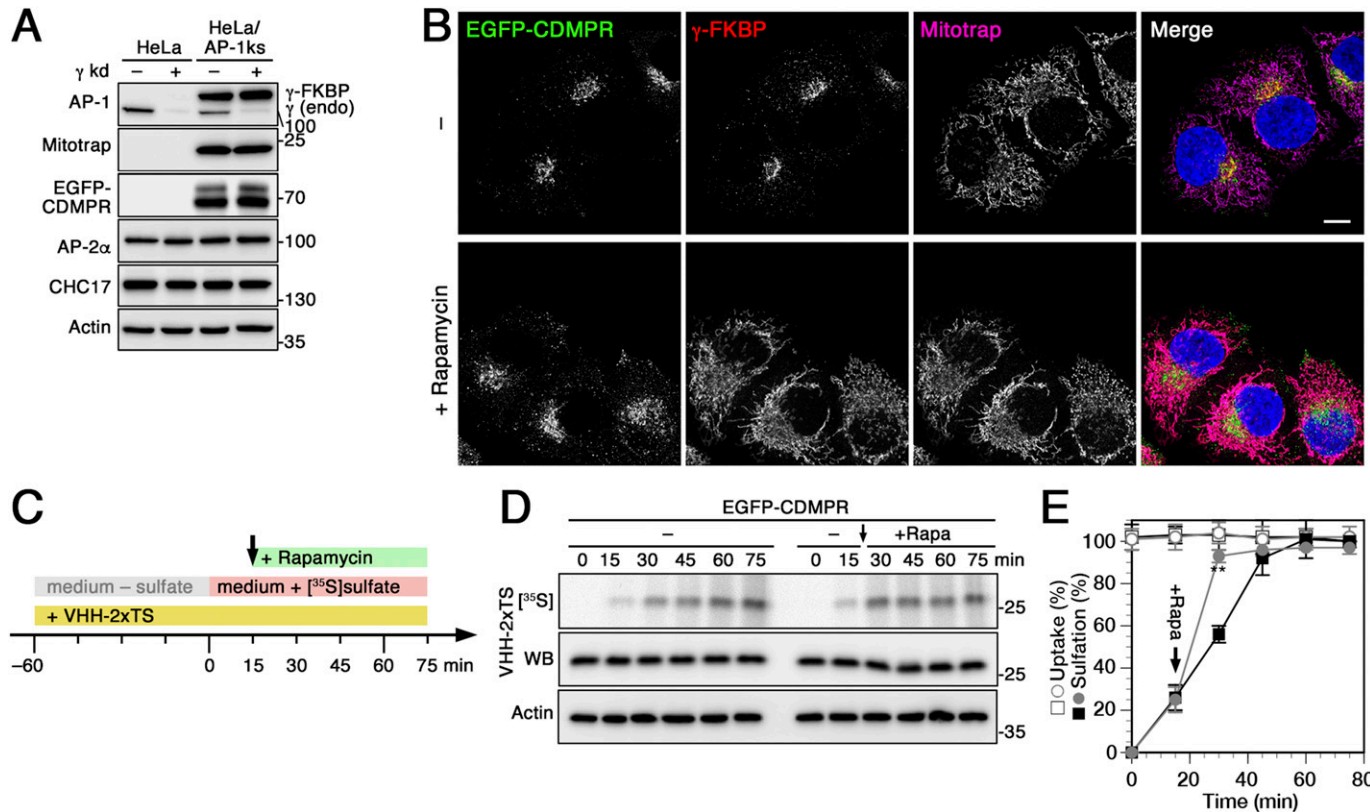

**Figure 4. Increased nanobody sulfation upon adaptor protein (AP)-1 silencing is the consequence of anterograde TGN exit block.**
**(A)** Lysates of normal HeLa cells and HeLa-AP1 knocksideways (HeLa-AP1ks) cells stably expressing γ-FKBP, Mitotrap and EGFP-CDMPR with or without siRNA-mediated knockdown of the endogenous γ-adaptin were subjected to immunoblot analysis for both forms of γ-adaptin, for Mitotrap (anti-FLAG), EGFP-CDMPR, the α-adaptin subunit of AP-2, and clathrin heavy-chain (CHC17). Knockdown efficiencies for endogenous γ-adaptin were typically >85%. **(B)** HeLa-AP1ks cells stably expressing EGFP-CDMPR after silencing endogenous γ-adaptin were treated with or without 500 nM rapamycin for 1 h and processed for fluorescence microscopy to detect EGFP-CDMPR, recombinant γ-FKBP (using an antibody targeting an epitope present in the neuronal splice variant of AP-2α), and Mitotrap (anti-FLAG). Bar: 10 μm. **(C)** Schematic outline of the anterograde transport sulfation assay in HeLa-AP1ks/EGFP-CDMPR cells. **(D)** HeLa-AP1ks cells stably expressing EGFP-CDMPR were siRNA-silenced for endogenous γ-adaptin, followed by starvation for sulfate in the presence of VHH-2xTS to preload all EGFP-CDMPR in the surface/endosome/TGN pool. The cells were then labeled with [³⁵S]sulfate for up to 75 min in the continued presence of VHH-2xTS, without or with addition of 500 nM rapamycin after 15 min (arrow) to inactivate AP-1 (+Rapa). The nanobodies were isolated by Ni/NTA beads and subjected to SDS-gel electrophoresis followed by immunoblot analysis (anti-His6) and autoradiography ([³⁵S]). In parallel, aliquots of the cell lysates were immunoblotted for actin as a control for the amount of cells used. **(E)** Three independent experiments as shown in panels C were quantified and presented as the percentage of the value in the absence of rapamycin after 75 min (mean and SD of three independent experiments; two-sided t test: *P < 0.05; **P < 0.01). Without rapamycin is shown as black squares, with rapamycin as gray circles; uptake as open symbols, sulfation as filled symbols.

transport of EGFP-CDMPR by ~50% after 75 min (Fig 5F and G), comparable with the effect of Vps26 depletion (Fig 2A and B). This result confirms a role of epsinR in endosome-to-TGN retrograde transport of CDMPR as previously suggested for CIMPR, TGN46, and STxB by Johannes and colleagues (Saint-Pol et al, 2004).

Surprisingly, however, depletion of the GGAs revealed a very similar and significant reduction of VHH-2xTS sulfation kinetics by ~40% with no apparent effect on uptake (Fig 5H and I). This result is not consistent with the expected unique function of GGAs in anterograde transport, but rather supports a role in retrograde traffic.

### Rapid GGA2 inactivation by knocksideways has no net effect on sulfation of CDMPR-bound nanobodies

If GGAs mediate anterograde transport of CDMPR out of the Golgi as indicated by previous studies, their silencing is expected to result in an increase in TGN residence time of CDMPR and imported

nanobodies and thus in an increase in sulfation. Surprised by the opposite finding upon gradual depletion of GGAs, we also tested the effect of rapid GGA inactivation using the well-characterized GGA2 knocksideways system by Robinson and colleagues (Hirst et al, 2012). Because their GGA2 knocksideways cell line expresses YFP-modified Mitotrap to anchor-away FKBP-tagged GGA2 adaptors, we replaced GFP by mCherry in the CDMPR reporter and used functionalized anti-mCherry nanobodies (LaM4; [Fridy et al, 2014]), constructed in the same way as the VHHGFP4 constructs (Fig 6A). The derivatized anti-mCherry nanobodies were produced at the same high purity, yield, and functionality (Fig 6B), and were specifically endocytosed by mCherry-tagged reporter proteins (shown for mCherry-CDMPR and transferrin receptor [TfR]-mCherry in Fig S4).

We stably expressed mCherry-CDMPR in the HeLaM-GGA2ks cells (Fig 6C). Rapamycin-triggered mitochondrial rerouting of FKBP-GGA2 adaptors for 1 h did not cause detectable changes in the

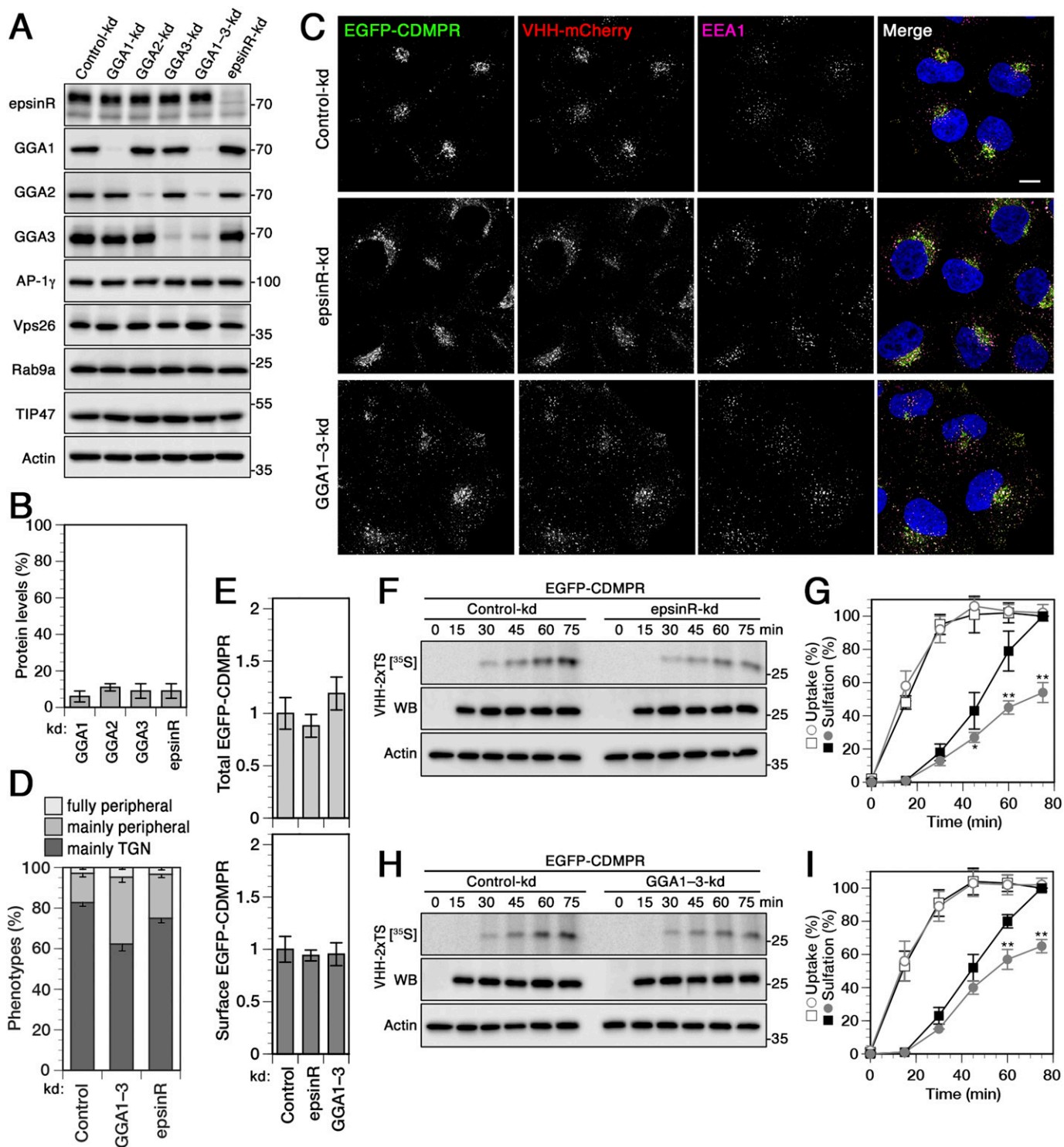

**Figure 5. Knockdown of the clathrin adaptors epsinR and GGA1–3 reduces retrograde transport of CDMPR.**
**(A)** HeLa cells were transfected with non-targeting siRNA (Control-kd) or siRNAs targeting GGA1, GGA2, and GGA3, individually or combined (GGA1–3-kd), or epsinR. 3 d after transfection, cells were analyzed by immunoblotting with antibodies against the indicated proteins. **(B)** To determine the knockdown efficiency, the residual protein was quantified in percent of the value after control-kd (mean and SD of three independent experiments). **(C)** HeLa cells stably expressing EGFP-CDMPR were depleted of epsinR, or all three GGAs (GGA1–3) as in (B). Cells were incubated for 1 h at 37°C with full medium containing 5 µg/ml VHH-mCherry (~0.1 µM), fixed, stained for EEA1 and nuclei (DAPI, blue), and imaged by fluorescence microscopy. Bar: 10 µm. **(D)** Quantitation of the percentage of cells displaying the CDMPR localization phenotypes "mainly TGN," "mainly peripheral," or "fully peripheral" (Wassmer et al, 2007; Simonetti et al, 2017). For each condition, random frames with a total of 137–152 cells were scored from three independent experiments. **(E)** Normalized levels of total and surface EGFP-CDMPR levels in RNAi-silenced cells were quantified by flow cytometry as in Fig 1G and H. Mean fluorescence intensities of each condition were normalized to the average of cells treated with non-targeting control siRNA. For each knockdown condition,

distribution of mCherry-CDMPR (Fig 6F). To ensure functionality of the modified GGA2 knocksideways cell line, we assayed missorting of myc-tagged procathepsin D precursor into the medium upon addition of rapamycin. Rerouting GGA2 to mitochondria resulted in a significant increase in procathepsin D secretion (Fig 6D and E).

To monitor the consequences of rapid GGA2 inactivation on retrograde CDMPR transport, cells were silenced for endogenous GGA1–3 by siRNA transfection, before they were incubated with LaM4-3xTS and simultaneously labeled with [$^{35}$S]sulfate in the presence or absence of rapamycin to rapidly remove the GGA2–FKBP fusion protein. Surprisingly, acute GGA2 depletion barely affected the sulfation kinetics of the nanobody imported by mCherry-CDMPR over 75 min (Fig 7A–C). This result is neither in agreement with GGAs specifically mediating retrograde, nor exclusively anterograde transport at the TGN–endosome interface. Rather, it could be the consequence of simultaneous reduction of transport in both directions.

To determine a potential direct role of GGA2 in TGN exit and thus anterograde transport, we applied the nanobody-preloading strategy (as already applied on AP-1ks cells in Fig 4C). We first loaded HeLa-GGA2ks cells expressing mCherry-CDMPR with LaM4-3xTS nanobody to steady-state during sulfate starvation, followed by [$^{35}$S]sulfate labeling for up to 75 min and where rapamycin was added 15 min post [$^{35}$S]sulfate addition (Fig 7D). Inactivation of GGA2 caused at best a mild, but not statistically significant increase in sulfation compared to cells treated with vehicle only (Fig 7E and F). Also this result would be consistent with compensatory effects on retrograde and anterograde transport of CDMPR of GGA2, notably in the absence of GGA1 and GGA3.

As proposed earlier for machinery depletion, the strategy of protein inactivation (gradual versus acute) turns out to be critical for phenotype analysis (Robinson et al, 2010; Hirst et al, 2012, 2015). Whereas knocksideways circumvent cellular adaptation and the up-regulation of potential compensatory mechanisms, long-term silencing by knockdown or knockout might be affected by accumulating indirect effects.

## Discussion

In the present study, we have systematically analyzed the contribution of various intracellular sorting machineries to retrieval of CDMPR from the cell surface to the TGN by a transport assay. Most of the earlier studies were based on the analysis of changes in steady-state distributions of transported proteins by fluorescence microscopy upon silencing of potential machinery components. To more directly and quantitatively assess transport and its kinetics, we took advantage of functionalized anti-GFP nanobodies in combination with cell lines stably expressing EGFP-CDMPR (Buser et al, 2018; Buser & Spiess, 2019). Nanobodies containing TS sites

report arrival and residence in the TGN, the compartment of sulfation, as they are internalized piggyback by EGFP-CDMPR from the cell surface. We previously applied this assay to test the contribution of AP-1 using the knocksideways system for rapid depletion (Buser et al, 2018). Because AP-1 is also implicated in anterograde transport from the TGN to endosomes, rapid inactivation promised less indirect effects resulting from inhibition of TGN exit than long-term silencing by siRNA-mediated knockdown or knockout. A clear reduction of the rate of sulfation by approximately one-third was detected after rapamycin-induced knocksideways, thus confirming a non-exclusive role of AP-1/clathrin in retrograde transport of CDMPR.

Here, we performed the experiment also with and without AP-1 inactivation by knockdown or knockout. The result was indeed strikingly different because hypersulfation was observed (Fig 3). Whereas other, AP-1–independent pathways still mediate significant retrograde transport of nanobody–EGFP-CDMPR complexes from endosomes to the TGN, their exit from the TGN is reduced by the absence of AP-1, extending their residence time in the sulfation compartment and thus the incorporation of [$^{35}$S] sulfate. We could show that pre-equilibrated nanobody–EGFP-CDMPR at the TGN was more strongly sulfated as soon as AP-1 depletion was triggered by rapamycin addition in knocksideways cells (Fig 4), clear evidence for inhibition of TGN exit and the anterograde role of AP-1-CCVs for CDMPR. Upon long-term depletion of AP-1, the steady-state pool of CDMPR at the TGN is likely higher than immediately after rapamycin-triggered knocksideways. This will further increase the residence time in the TGN and thus sulfation of entering nanobody–EGFP-CDMPR complexes and may account for the strong increase in sulfation after knockdown and knocksideways that overcompensates the reduction in incoming CDMPR.

It cannot be excluded that additional indirect effects caused by gradual and long-term depletion contribute to hypersulfation. Several unexpected and unexplained phenomena have previously been observed upon knockdown of AP-1, but not upon knocksideways: almost no reduction of CIMPR and ARF1 in CCVs, but increased AP-2 levels (Robinson et al, 2010; Navarro Negredo et al, 2017), and GGA2 was still incorporated into CCVs isolated from AP-1 knockdown but not from knocksideways cells (Hirst et al, 2012). However, tyrosine sulfation activity was not affected in AP-1 knockdown cells (Fig S2). Interestingly, in a proteomics search for CCV content dependent on AP-1, SLC35B2, one of the transporters delivering the activated sulfation precursor 3′-phosphoadenosine-5′-phosphosulfate (PAPS) into the TGN lumen scored positive (Hirst et al, 2012). One might thus speculate that lack of retrieval of this and other components of the sulfation machinery specifically by AP-1 might lead to hypersulfation in endosomes. Our observation, however, that localization of Golgi sulfotransferases remained largely unaffected makes this scenario appear unlikely, in particular

50,000 cells were analyzed in each experiment (mean and SD of three independent experiments). **(F, G, H, I)** Cells stably expressing EGFP-CDMPR were transfected with non-targeting siRNA (control-kd) or with siRNA silencing expression of epsinR (F, G) or GGA1–3 (H, I) as described in (A). The cells were labeled with [$^{35}$S]sulfate for up to 75 min in the presence of 2 μg/ml VHH-2xTS and the nanobodies were isolated, analyzed, and quantified as in Fig 2 (mean and SD of three independent experiments; two-sided t test: *P < 0.05; **P < 0.01). Control-kd is shown as black squares and epsinR/GGA1–3-kd as gray circles; uptake as open symbols, sulfation as filled symbols.

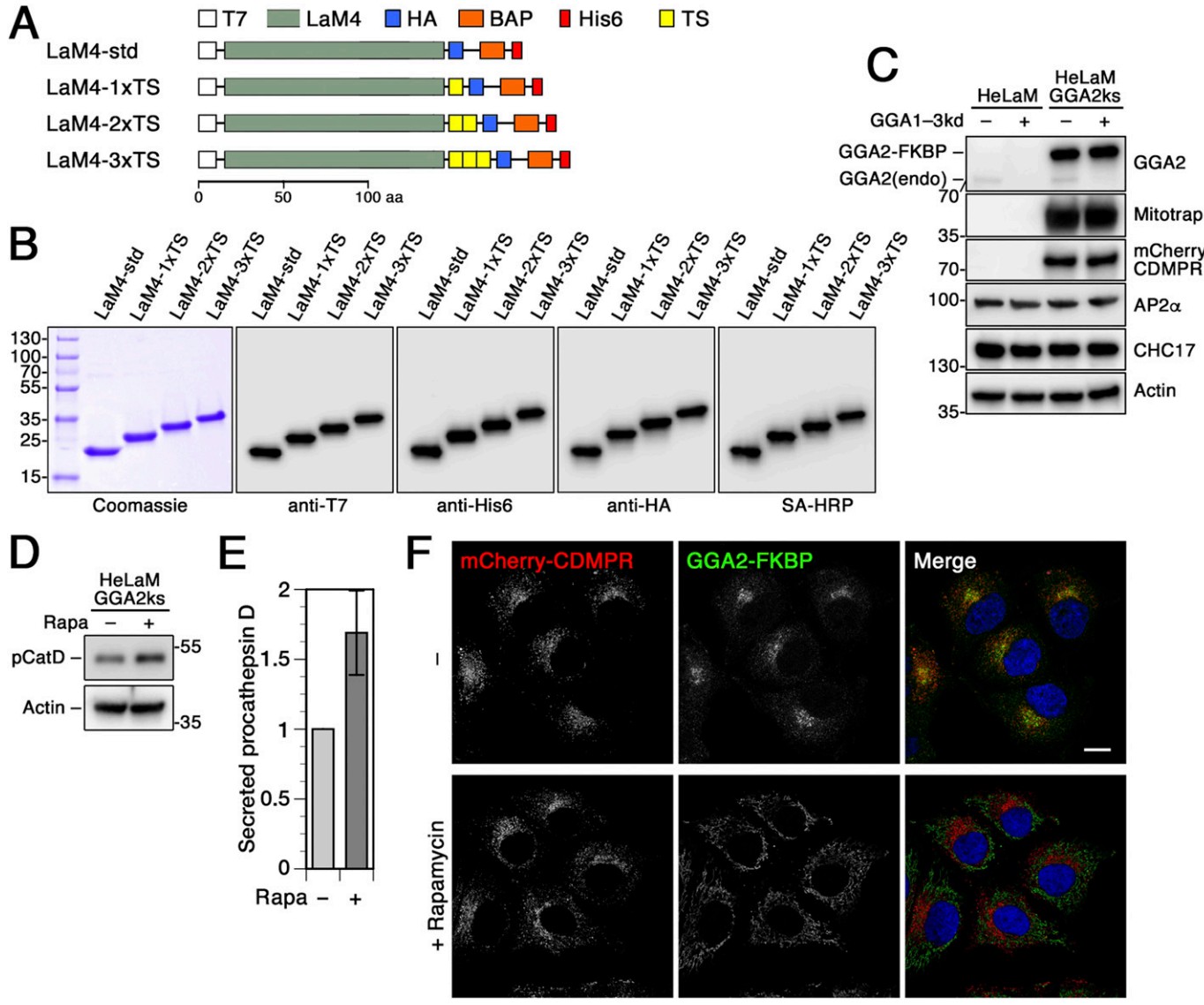

**Figure 6. Derivatized anti-mCherry nanobodies for use with GGA2 knocksideways cells.**
**(A)** Schematic representation of the functionalized anti-mCherry nanobodies. The standard nanobody (LaM4-std) consists of the mCherry-specific LaM4 domain, T7 and HA epitope tags, a biotin acceptor peptide (BAP), and a hexahistidine (His6) purification tag. Other nanobodies in addition contain one to three tyrosine sulfation sequences (TS). Scale bar in aa. **(B)** Bacterially expressed and purified nanobodies (30 μg) were analyzed by SDS-gel electrophoresis and Coomassie staining (left). Immunoblot analysis of nanobodies (10 ng) with antibodies against the HA, His6, or T7, or with streptavidin-HRP (SA-HRP). Marker proteins with molecular weights in kilodalton are shown on the left. **(C)** Parental HeLaM cells and HeLaM-GGA2ks cells stably expressing mCherry-CDMPR were transfected with non-targeting siRNA (–) or siRNAs silencing endogenous GGA1–3 (+). Cell lysates were subjected to immunoblot analysis with antibodies against the indicated proteins. **(D)** HeLaM-GGA2ks cells stably expressing mCherry-CDMPR were transfected with siRNAs targeting endogenous GGA1–3. These cells were transfected with a plasmid expressing His6/myc-tagged procathepsin D 24–36 h before analysis. Media of cells incubated for 2 h in serum-free medium supplemented with 5 mM mannose-6-phosphate to prevent cathepsin D binding to surface MPRs, and with or without rapamycin (+ or – Rapa, respectively) were analyzed by collecting procathepsin D (pCatD) with Ni/NTA beads and immunoblotting with anti-myc antibodies. Cell lysates were immunoblotted for actin as a control. **(E)** Procathepsin D missorted upon knocksideways of GGA2 (+Rapa) was quantified from immunoblots as shown in panel (D), normalized to the DMSO-treated (–Rapa) control (mean and SD of four independent experiments). **(F)** HeLa-GGA2ks cells stably expressing mCherry-CDMPR after silencing endogenous GGA1–3 were treated with or without 500 nM rapamycin for 1 h and processed for fluorescence microscopy to detect mCherry-CDMPR and recombinant GGA2-FKBP. Bar: 10 μm.

because a concomitant redistribution of sulfotransferases and PAPS transporters is required for proper sulfate incorporation in proteins (Dick et al, 2008). Irrespective of that, one should mention that Johannes and colleagues observed a slight increase in STxB sulfation, when AP-1 was silenced by siRNA (Saint-Pol et al, 2004). Their findings with AP-1 have not been commented.

In any case, using our assay, it is expected that silencing of components involved in retrograde transport machineries for CDMPR causes reduced rates of nanobody sulfation, and depletion of proteins mediating TGN exit causes increased rates of sulfation. Accordingly, we found a clear reduction of nanobody sulfation upon knockdown of Vps26, confirming the role of retromer in retrograde

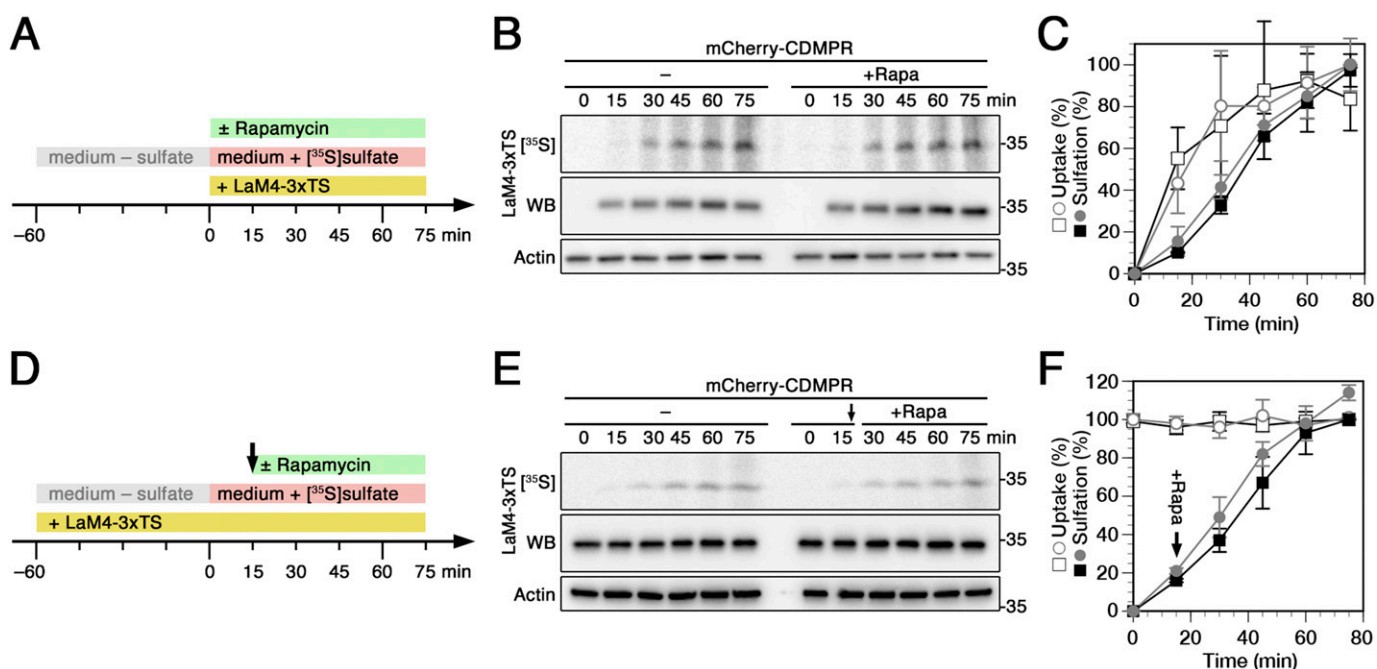

**Figure 7. Effect of rapid GGA2 inactivation on retrograde and anterograde CDMPR transport using derivatized anti-mCherry nanobodies.**
**(A, D)** Schematic outline of the retrograde (A) anterograde (D) transport sulfation assays, respectively, with HeLa-GGA2ks/mCherry-CDMPR cells. **(B)** HeLa-GGA2ks cells stably expressing mCherry-CDMPR were siRNA-silenced for endogenous GGA1–3. The cells were starved for sulfate for 1 h, and then labeled for up to 75 min with [35S] sulfate in the presence of LaM4-3xTS nanobodies either in the absence (–) or presence of 500 nM rapamycin to inactivate GGA2-FKBP (+Rapa). The nanobodies were isolated by Ni/NTA beads and subjected to SDS-gel electrophoresis followed by immunoblot analysis (anti-His6) and autoradiography ([35S]). In parallel, aliquots of the cell lysates were immunoblotted for actin as a control for the amount of cells used. **(C)** Experiments as shown in panel B were quantified and presented as the percentage of the value in the absence of rapamycin after 75 min (mean and SD of three independent experiments). Without rapamycin is shown as black squares, with rapamycin as gray circles; uptake as open symbols, sulfation as filled symbols. **(E)** HeLa-GGA2ks cells stably expressing mCherry-CDMPR were siRNA-silenced for endogenous GGA1–3, followed by starvation for sulfate in the presence of LaM4-3xTS to preload mCherry-CDMPR in the surface/endosome/TGN pool. The cells were then labeled with [35S] sulfate for up to 75 min in the continued presence of LaM4-3xTS, without or with addition of 500 nM rapamycin after 15 min (arrow) to inactivate GGA2 (+Rapa). (B) Nanobody analysis was performed as above (B). **(F)** Experiments as shown in panel E were quantified and presented as the percentage of the value in the absence of rapamycin after 75 min (mean and SD of three independent experiments). Without rapamycin is shown as black squares, with rapamycin as gray circles; uptake as open symbols, sulfation as filled symbols.

transport also for CDMPR. There are many reports for a requirement of retromer for CIMPR (Arighi et al, 2004; Seaman, 2004, 2007, 2018; Wassmer et al, 2007; Bulankina et al, 2009; Seaman et al, 2009; Harbour et al, 2010; Bugarcic et al, 2011; Fjorback et al, 2012; Hao et al, 2013; McGough et al, 2014; Hirst et al, 2018; Wang et al, 2018; Chen et al, 2019; Cui et al, 2019). However, the requirement of the trimeric Vps retromer complex for CIMPR retrieval has recently been challenged by reports of the Cullen and Steinberg labs (Kvainickas et al, 2017; Simonetti et al, 2017, 2019; Evans et al, 2020). Using colocalization analyses, they failed to detect mislocalized CIMPR in Vps35-inactivated cells by knockdown and knocksideways, but identified a specific motif in CIMPR's cytosolic tail (WLM, not present in CDMPR) that binds to the SNX-BARs SNX1/2 and SNX5/6 and is required for correct receptor localization. CI- and CDMPRs may well differ in their interactions with retromer components. Furthermore, loss of either Vps26 or Vps35, two components of the retromer core complex, produced very different phenotypes for retrograde STxB transport to the TGN (Popoff et al, 2009), showing potential alternate functions of the retromer subunits.

Defects in retrograde transport of CDMPR characteristically goes together with a change in steady-state distribution of the receptor in favor of peripheral endosomal compartments. This was found to be the case, when Vps26 or AP-1 was silenced, but not detectably for TIP47 and only to a small extent for Rab9a (Fig 1). In agreement with this result, no change in nanobody sulfation was observed upon TIP47 knockdown and a considerable reduction upon Rab9a silencing (Fig 2). Our results thus do not confirm a role of TIP47 as a sorting device for CDMPR transport in vivo, but show an impact of Rab9a depletion on CDMPR arrival in the TGN. The effect of Rab9a depletion on retrograde MPR transport might also be to general defects in endosomal maturation. Rab9a knocksideways could indeed be helpful to assess directly the effects on CDMPR retrograde transport.

Silencing epsinR was found to cause a clear reduction of nanobody sulfation and a slight redistribution of CDMPR to the periphery (Fig 5), consistent with a role of epsinR in retrograde transport of CDMPR. This result adds to previous studies showing a function of epsinR on distribution or transport of CIMPR and STxB (Saint-Pol et al, 2004).

Most surprising was our finding that RNAi-mediated depletion of GGA1-3 did not produce hypersulfation of nanobodies imported by EGFP-CDMPR as expected for a component involved in TGN exit, but rather reduced sulfation indicative of a defect in retrograde transport (Fig 5). Consistent with this notion, significant peripheral

redistribution of CDMPR was observed and not perinuclear accumulation at the TGN. The results contradict an exclusive role of GGAs in anterograde transport of MPRs in cooperation with AP-1 as previously proposed (Doray et al, 2002, 2021; Ghosh et al, 2003a; Ghosh & Kornfeld, 2004). Contrary to the knockdown experiments silencing all three GGAs, rapid depletion of overexpressed GGA2 in the absence of GGA1 and GGA3 by knocksideways had no significant effect on the sulfation kinetics of CDMPR-imported nanobodies, neither hypersulfation supporting reduced TGN exit, nor hyposulfation consistent with reduced retrograde transport (Fig 7). This result suggests either that GGA2 contributes to neither anterograde nor retrograde transport of CDMPR, or that it is involved in both directions, whereby the opposing effects on nanobody sulfation could cancel each other out. In the latter case, one would have to assume that long-term depletion of the GGAs in the knockdown experiment produced changes/adaptations to specifically compensate the TGN exit defect. To more directly assess a role of GGA2 in anterograde transport, we imported nanobody into steady state before sulfation and acute inactivation. Rapid depletion of GGA2 caused a mild increase in sulfation consistent with a contribution to TGN exit and with the hypothesis of bidirectional functions. Direct comparison of knockdown experiments for all three GGAs with GGA2 knocksideways experiments in the absence of GGA1 and GGA3 is valid only under the assumption of complete redundancy between all three GGAs. This has not been analyzed in detail yet.

GGAs clearly localize both to the TGN and to endosomes (Dell'Angelica et al, 2000; Hirst et al, 2000; Ghosh et al, 2003b; Wahle et al, 2005; D'Souza et al, 2014; Ratcliffe et al, 2016; Uemura et al, 2018; Uemura & Waguri, 2020), like AP-1, and thus might operate at both places. Because GGA depletion had previously been observed to cause redistribution of CIMPR to EEA1-positive compartments, a role also in retrograde transport from endosomes had not been excluded (Ghosh et al, 2003b). Comparative CCV proteomics with GGA2 and AP-1 knocksideways cells pointed towards involvement of GGA/AP-1 coats in anterograde sorting of MPR–lysosomal hydrolase complexes from the TGN (Hirst et al, 2012), yet the authors did not discount a potential retrograde function. A very surprising finding by Hirst et al (2012) was that Rabaptin5 (RABEP1) was the only known accessory component to be significantly lost from GGA2 knocksideways CCVs. Rabaptin5 is a marker of early endosomes where, as a complex with Rabex5 it activates Rab5 (Kalin et al, 2016). The fact that GGA depletion affects the CCV association of an endosomal protein points towards a role of these adaptors on endosomes, possibly in retrograde transport.

Interestingly, a recent study performed in *Schizosaccharomyces pombe* demonstrated that GGAs in collaboration with clathrin adaptors indeed contribute to efficient retrograde transport of Vps10, yeast's MPR homologue, from the prevacuolar endosome to the TGN (Yanguas et al, 2019). In addition, despite the functional relationship of GGA2 and AP-1 adaptors, it is surprising to observe that they are spatially segregated from each other to a considerable extent (Huang et al, 2019). Our results showing reduced CDMPR transport to the TGN using a sulfation-based approach strongly support a contribution to retrograde transport by GGAs.

In the present study, we have analyzed the contribution of individual sorting machineries in retrograde endosome-to-TGN transport of CDMPR and found that several machineries contribute, likely from

different types of endosomes: retromer, the clathrin adaptors AP-1, epsinR, and—most surprisingly—the GGAs, and Rab9a. Other sorting machinery components that might facilitate endosome-to-TGN transport of MPRs include Rab29, Rab35, SNX-BAR proteins, and AP-5 (Wang et al, 2014; Cauvin et al, 2016; Kvainickas et al, 2017; Simonetti et al, 2017, 2019; Hirst et al, 2018). The two latter machineries have been shown to be important for CIMPR retrieval, not yet for CDMPR. Future systematic and comparative analyses of machinery requirement between CDMPR and CIMPR in retrograde transport for these machineries are important to further understand the coexistence and operation of multiple TGN retrieval pathways for receptors. Although CDMPR and CIMPR have completely different cytosolic tails and sorting motifs, they seem to largely share the same sorting machineries for their transport.

Importantly, our study also highlights the values and caveats of different depletion assays. RNAi silencing may be incomplete and may thus not reveal the full phenotypes. Protein depletion is gradual and thus allows compensatory mechanisms to be activated. Whereas in knockout cells, the protein of interest is completely absent, in both systems the long-term lack of a function may lead to indirect effects, such as the mislocalization of machinery components (e.g., SNAREs) involved in other transport steps. These disadvantages are avoided by rapid depletion by knocksideways, which aims to surprise the cell by the sudden absence of a protein. It requires, however, expression of a fusion protein that is fully functional and ideally expressed at a similar level as the original protein. Completeness of removal upon rapamycin addition is difficult to assess (particularly when highly overexpressed). In the case of the GGAs, the situation is complicated the existence of three isoforms. Their complete redundancy is not established, since depletion of single GGAs already showed effects on cathepsin processing and sorting of (Ghosh et al, 2003a; Mardones et al, 2007). The available knocksideways cell line specifically addresses the effect of rapid depletion of overexpressed GGA2-FKBP in cells lacking GGA1 and GGA3.

The use of sulfatable nanobodies provides a new perspective on the sorting machineries of CDMPR. Our study highlights the critical differences of inactivation strategies (knockout, knockdown, knocksideways), but also provides evidence for a role of GGAs in retrograde transport of CDMPR, consistent with their localization to endosomes.

## Materials and Methods

### Bacterial expression and purification of functionalized nanobodies

Functionalized nanobodies were bacterially expressed and isolated as previously described (Buser et al, 2018; Buser & Spiess, 2019). Briefly, bacterial expression vectors encoding derivatized VHH or LaM4 nanobodies and myc-BirA (#109424; Addgene) were transformed together into Rosetta DE3 cells (Merck), and plated on LB plates with 50 $\mu$g/ml kanamycin and 50 $\mu$g/ml carbenicillin. A 20-ml overnight culture of a single colony was diluted into 1 liter LB medium with antibiotics and 200 $\mu$M D-biotin and grown to an $OD_{600}$ of 0.6–0.7 at 37°C. Expression was induced with 1 mM isopropyl-$\beta$-D-

thiogalactopyranoside (IPTG) at 16°C overnight (VHH-mCherry and LaM4-EGFP), or at 30°C for 4 h (VHH-std, VHH-2xTS, LaM4-std, LaM4-1xTS, LaM4-2xTS, and LaM4-3xTS). Cells were pelleted at 5,000*g* at 4°C for 45 min and stored at –80°C. Upon thawing, they were resuspended in 30 ml PBS with 20 mM imidazole, 200 $\mu$g/ml lysozyme, 20 $\mu$g/ml DNase I, 1 mM MgCl$_2$, and 1 mM PMSF, incubated for 10 min at room temperature and 1 h at 4°C while rotating, followed by mechanical lysis using a tip sonicator for three times 30 s with 1-min cooling periods. The lysate was cleared by centrifugation at 15,000*g* for 1 h at 4°C and loaded on a His GraviTrap column (GE Healthcare Life Sciences), washed with 20 mM imidazole in PBS, and eluted with 2 ml PBS with 500 mM imidazole. The purified nanobodies were desalted on PD-10 columns (GE Healthcare Life Sciences), concentrated to 2 mg/ml (VHH-std, VHH-2xTS, LaM4-std, LaM4-1xTS, LaM4-2xTS, and LaM4-3xTS) or 5 mg/ml (for VHH-mCherry and LaM4-EGFP), flash-frozen in liquid nitrogen, and stored at –80°C. Plasmids for nanobody fusion protein expression are deposited with Addgene (#109417, VHH-std; #109419, VHH-2xTS; #109421, VHH-mCherry; Addgene). TS- und EGFP-modified LaM4 derivatives will be made available by Addgene (#162777–162779, #182641; Addgene).

### Cell culture, plasmids, CRISPR/Cas9 gene editing, and RNA interference

HeLa $\alpha$ cell lines were maintained in high-glucose DMEM with 10% FCS, 100 units/ml streptomycin, 2 mM L-glutamine and appropriate selection antibiotics (1.5 $\mu$g/ml puromycin, 1 mg/ml hygromycin B, or 7.5 $\mu$g/ml blasticidin) at 37°C in 7.5% CO$_2$. The HeLa $\alpha$ cell line was cell line authenticated and proven to be human (Microsynth). Phoenix Ampho packaging cells (from the Nolan laboratories, Stanford University) were grown in complete medium supplemented with 1 mM sodium pyruvate.

HeLa cells stably expressing EGFP-CDMPR, and HeLa-AP1ks cells stably expressing the respective EGFP reporter were previously described (Buser et al, 2018). HeLaM-GGA2ks cells were a generous gift of Scottie Robinson and Jennifer Hirst.

To generate HeLa cells stably expressing SHMY-A1Pi, SHMY-PAUF, cathepsin D-YMH, TPST1-EGFP, and TPST2-EGFP, Phoenix Ampho packaging cells were transfected pQCXIP-SHMY-A1Pi, pQCXIP-SHMY-PAUF, pQCXIP-cathepsin D-YMH, pQCXIP-TPST1-EGFP, and pQCXIP-TPST2-EGFP using FuGENE HD (Promega). The viral supernatant was harvested after 48 h, passed through a 0.45 $\mu$m filter, supplemented with 15 $\mu$g/ml polybrene, and added to target HeLa $\alpha$ cells. The next day, complete medium with 1.5 $\mu$g/ml puromycin was added for selection, and pooled resistant clones were used for experiments. HeLa cells expressing TPST1-EGFP or TPST2-EGFP were further subjected to cell sorting on a FACSAria III (BD Biosciences) to obtain a cell pool with homogeneous expression levels. cDNA sequences of PAUF or cathepsin D were generous gifts by Vivek Malhotra (CRG) or Stuart Kornfeld (WUSM). Reporter plasmids were deposited on Addgene (#182642, EGFP-CDMPR; #182643, mCherry-CDMPR; #182644, cathepsin D-YMH; #182645, SHMY-PAUF; #182648, SHMY-A1Pi; Addgene).

A plasmid encoding TPST1-EGFP (#66617; Addgene) or TPST1-EGFP (#66618; Addgene) were kind gifts from David Stephens. The anti-mCherry nanobody sequence (LaM4) based on Rout and colleagues

(Fridy et al, 2014) was a kind gift from Kazuhisa Nakayama (#70696; Addgene).

Generation of HeLaM-GGA2ks expressing mCherry-CDMPR (in pQCXIP) were established as outlined above, but selected and propagated with 1.5 $\mu$g/ml puromycin, 500 $\mu$g/ml hygromycin B, and 500 $\mu$g/ml G418. Homogeneous mCherry-CDMPR expression was obtained by cell sorting.

To generate a $\gamma$-adaptin knockout HeLa cell line by CRISPR/Cas9, sgRNAs for gene editing were purchased from Santa Cruz Biotechnology (sc-403986). Briefly, parental HeLa cells were transfected with 2 $\mu$g plasmid containing a GFP cassette and 4 $\mu$l FuGENE HD (Promega) in a six-well cluster. After 24 h of expression, cells were subjected to FACS, and single cells or pooled clones were collected. Transient transfection of AP-1ko cells with pQCXIP-$\gamma$-FKBP, pQCXIP-TPST1-EGFP, or pQCXIP-TPST2-EGFP was performed using FuGENE HD according to the manufacturer's instructions. Anti-CIMPR antibody uptake was performed as described previously (Robinson et al, 2010).

For RNA interference, cells were reverse-transfected with target siRNA in Opti-MEM I using Lipofectamine RNAiMAX (both Thermo Fisher Scientific) following the manufacturer's instructions. For a conventional knockdown of AP-1, the sequence 5′-AAGGCAUCAA-GUAUCGGAAGAdTdT-3′ against the $\mu$1A-subunit of the heterotetrameric complex was used as formerly reported (Hirst et al, 2003, 2005, 2009). For RNA interference with retromer complex, we applied siRNA duplexes with the sequence 5′-AACUCCUGUAACCCUU-GAGdTdT-3′ targeting Vps26 as described in previous studies (Popoff et al, 2007, 2009). To specifically silence Rab9 or TIP47, we applied the siRNA sequence 5′-GUUUGAUACCCAGCUCUUCdTdT3′ for Rab9 (Ganley et al, 2004; Reddy et al, 2006; Kucera et al, 2016), or 5′-CCCGGGGCUCAUUUCAAACdTdT-3′ for TIP47 (Bulankina et al, 2009). EpsinR was targeted with 5′-AAUACAGAUAUGGUCCAGAAUTdTdT-3′, GGA1 with 5′-CACAGGAGUGGGAGGCGAUUTdTdT-3′, GGA2 with 5′-UGAAUUAUGUUUCGCAGAAUTdTdT-3′, and GGA3 with 5′-UGUGA-CAGCCUACGAUAAAUTdTdT-3′ as previously described (Hirst et al, 2004, 2012). To knockdown AP-2$\alpha$ and CHC17, we used 5′-AA GAGCAUGUGCACGCUGGCCAdTdT-3′ and 5′-UAAUCCAAUUCGAAGA CCAAUdTdT-3′ duplexes, respectively, as previously described (Motley et al, 2003). We used the non-targeting siRNA 5′-UA AGGCUAUGAAGAGAUACdTdT-3′ as control siRNA (Salazar et al, 2009). All siRNAs were used at a final concentration of 50 nM, apart from the GGA siRNAs (used at 25 nM each).

To silence $\gamma$-adaptin in HeLa-AP1ks cells, the siRNA sequence 5′-GAAGAUAGAAUUCACCUUUUU-3′ was used as previously described (Robinson et al, 2010; Buser et al, 2018). Cells were transfected twice (day 1 and 3) and used at day 5. siRNA duplexes were purchased from Microsynth.

### Gel electrophoresis and immunoblot analysis

Proteins separated by SDS-gel electrophoresis (7.5–15% polyacrylamide) were transferred to Immobilon-P$^{SQ}$ PVDF membranes (Millipore). After blocking with 5% non-fat dry milk in TBS (50 mM Tris·HCl, pH 7.6, 150 mM NaCl) with 0.1% Tween-20 (TBST) for 1 h, the membranes were probed with primary antibodies in 1% BSA in TBST for 2 h at room temperature or overnight at 4°C, followed by incubation with HRP-coupled secondary antibodies in 1% BSA in TBST

for 1 h at room temperature. Immobilon Western Chemiluminescent HRP Substrate (Millipore) was used for detection, a Fusion Vilber Lourmat Imaging System for imaging, and Fiji software for quantitation.

### Secretion assay

To monitor effects of machinery depletion on CDMPR-mediated lysosomal sorting, HeLa cells stably expressing cathepsin D-YMH or SHMY-PAUF were reverse-transfected with target siRNA as outlined above. After RNAi, cells were washed and incubated in serum-free medium for 1 h in the presence of 5 mM mannose-6-phosphate to avoid subsequent endocytosis of the secreted cathepsin precursor. Precipitation of recombinant His6-tagged cathepsin precursors from the medium was performed with Ni Sepharose High Performance beads (GE Healthcare Life Sciences) for 1 h at 4°C. Beads were washed three times with lysis buffer containing 20 mM imidazole and boiled in SDS-sample buffer. In addition, a fraction (50–100 $\mu$l) of the cell lysate was used for an actin control. Samples were run on an SDS–PAGE and analyzed by immunoblotting.

To assay recombinant cathepsin secretion in RNAi-silenced HeLaM-GGA2ks cells, cells were transiently transfected with pQCXIP-cathepsin D-YMH 24–36 h before incubation with serum-free medium supplemented with 5 mM mannose-6-phosphate in the presence or absence of 500 nM rapamycin. Secretion was biochemically assessed as outlined above.

### Fluorescence microscopy

For immunofluorescence staining, cells were grown on glass coverslips, fixed with 3% PFA for 10 min at room temperature, washed with PBS, quenched with 50 mM NH$_4$Cl in PBS for 5 min, permeabilized with 0.1% Triton X-100 in PBS for 10 min, blocked with 1% BSA in PBS for at least 15 min, incubated with primary antibody in BSA/PBS for 2 h, washed, and stained with fluorescent secondary antibodies in BSA/PBS for 1 h. After a 5 min staining with 5 $\mu$g/ml DAPI and three washes with PBS, coverslips were mounted in Fluoromount-G (Southern Biotech). Staining patterns were imaged on a Zeiss Point Scanning Confocal LSM700 or LSM880 microscope in super-resolution mode with the Airyscan detector. Receptor mislocalization analysis was performed by scoring and quantification of the percentage of cells displaying each phenotype in machinery-depleted cells using a Zeiss Axioplan microscope with a Leica DFC420C imaging system as described previously (Wassmer et al, 2007; Simonetti et al, 2017).

### Flow cytometry

To quantitatively assess changes in total and surface abundance of recombinant CDMPR upon silencing of machinery components, parental HeLa cells as well as HeLa cells stably expressing EGFP-CDMPR, untreated or after treatment with non-targeting control siRNA or various targeting siRNAs, were incubated with 5 $\mu$g/ml purified VHH-mCherry in complete medium for 30 min at 4°C to specifically bind surface EGFP-CDMPR. After six washes with ice-cold medium to remove unbound nanobody, cells were gently harvested with non-enzymatic cell dissociation solution (Millipore).

Collected cells were resuspended in ice-cold PBS with 0.5% FBS for flow cytometry analysis using a LSR Fortessa Analyzer (BD Biosciences) with a blue (488 nm) and yellow-green (561 nm) laser. Surface CDMPR (mCherry) and total receptor (GFP) levels above the background of parental HeLa cells were quantified, as median fluorescence intensities using FCS Express Research 7 (De Novo Software) and FlowJo 10 (BD), and normalized to the mean value of control knockdown cells.

### Sulfation analysis

To analyze retrograde transport and kinetics of EGFP-CDMPR to the compartment of sulfation, cells were incubated with 1 ml sulfate-free medium for 1 h at 37°C and 7.5% CO$_2$ before labeling with sulfate-free medium supplemented with 0.5 mCi/ml [$^{35}$S]sulfate (Hartmann Analytics) and 2 $\mu$g/ml purified VHH-2xTS for up to 75 min. For the knocksideways experiment, HeLa-AP1ks or HeLaM-GGA2ks cells stably expressing EGFP-CDMPR or mCherry-CDMPR, respectively, were starved with sulfate-free medium for 1 h at 37°C and 7.5% CO$_2$ in the presence of 2 $\mu$g/ml purified VHH-2xTS or LaM4-3xTS, followed by two 1 ml washes with sulfate-free medium, before labeling with medium reconstituted with 0.5 mCi/ml [$^{35}$S]sulfate. 500 nM rapamycin from a 2,000× stock solution in DMSO, or DMSO alone was added. Cells stably expressing SHMY-A1Pi were starved as described above and pulsed with medium reconstituted with 0.5 mCi/ml [$^{35}$S]sulfate for 75 min.

After incubation, cells were washed twice with ice-cold PBS, lysed in 1 ml lysis buffer containing 2 mM PMSF and protease inhibitor cocktail, and centrifuged at 10,000$g$ for 15 min at 4°C. A fraction (50–100 $\mu$l) of the postnuclear supernatants was used for immunoblot analysis of total cell-associated nanobody and an actin control. The rest was incubated for 1 h at 4°C with 20 $\mu$l Ni Sepharose High Performance beads (GE Healthcare Life Sciences) to isolate the nanobodies. Beads were washed three times with lysis buffer containing 20 mM imidazole and boiled in SDS-sample buffer. Nanobodies were analyzed by SDS-gel electrophoresis and autoradiography using BAS Storage Phosphor Screens and a Typhoon FLA7000 IP phosphorimager (GE Healthcare Life Sciences).

### Antibodies

For immunofluorescence microscopy, goat anti-$\alpha$-adaptin (#EB11875; 1:1,000; Everest Biotech), mouse anti-CIMPR (ab2733; 1:5,000; Abcam), anti-EEA1 (#610456; 1:1,000; BD Biosciences), rabbit anti-FLAG (#2368; 1:500; Cell Signaling Technology), and rabbit anti-GM130 (#12480; 1:1,000; Cell Signaling Technology) antibodies were used.

For immunoblotting, mouse anti–$\alpha$-adaptin (#610501; 1:5,000; BD Biosciences), mouse anti–$\gamma$-adaptin (#610385; 1:5,000; BD Biosciences), mouse anti–$\beta$1/2-adaptin (#610381; 1:5,000; BD Biosciences), mouse anti–$\gamma$-adaptin (made from 100/3 hybridoma; 1:5,000), rabbit anti–$\sigma$1-adaptin (#A305-396A-M; 1:1,000; Bethyl Laboratories), mouse anti–$\gamma$-adaptin (made from 100/3 hybridoma; 1:5,000), mouse anti-actin (#MAB1501; 1:100,000; EMD Millipore), mouse anti-CHC17 (made from TD.1 hybridoma; 1:200), rabbit anti-epsinR (#A301-926A; 1:1,000; Bethyl Laboratories), mouse anti-FLAG (#8146; 1:1,000; Cell Signaling Technology), mouse anti-GFP (#11814460001-Roche;

1:5,000; Sigma-Aldrich), rabbit anti-GGA1 (#A305-368A; 1:1,000; Bethyl Laboratories), mouse anti-GGA2 (#612612; 1:2,000; BD Biosciences), mouse anti-GGA3 (#612310; 1:1,000; BD Biosciences), mouse anti-HA (made from 12CA5 hybridoma; 1:10,000), rabbit anti-His6 (#A190-114A; 1:10,000; Bethyl Laboratories), rabbit anti-mCherry (#GTX128508; GeneTex or #PA5-3497; Thermo Fisher Scientific; 1:10,000), rabbit anti-myc (#GTX29106; GeneTex), rabbit anti-T7 (#A190-117A; 1:10,000; Bethyl Laboratories), rabbit anti-Rab9a (#5118; 1:1,000; Cell Signaling Technology), mouse anti-SNX1 (#611482; 1:500; BD Biosciences), rabbit anti-SNX2 (#A304-544A; 1:2,000; Bethyl Laboratories), rabbit anti-TIP47 (10694-1-AP; 1:1,000; Proteintech), rabbit anti-Vps26 (#A304-801A; 1:1,000; Bethyl Laboratories), rabbit anti-Vps35 (#A304-727A; 1:1,000; Bethyl Laboratories), and mouse anti-$\mu$1A (H00008907-A01; 1:1,000; Abnova) antibodies were used.

As secondary antibodies for immunofluorescence microscopy, A568-labeled donkey anti-goat (#A-11057; 1:500; Thermo Fisher Scientific), A647-labeled donkey anti-goat (#A-21447; 1:500; Thermo Fisher Scientific), A633-labeled goat anti-mouse (#A-21052; 1:500; Thermo Fisher Scientific), A633-labeled goat anti-rabbit (#A-21071; 1:500; Thermo Fisher Scientific) immunoglobulin antibodies were used. As secondary antibodies for immunoblotting, HRP-labeled goat anti-rabbit (#A-0545; 1:10,000; Sigma-Aldrich), and goat anti-mouse (#A-0168; 1:10,000; Sigma-Aldrich) immunoglobulin antibodies were used. To detect biotinylated proteins on blots, Streptavidin-HRP (#434323; 1:10,000; Thermo Fisher Scientific) was used.

## Supplementary Information

## Acknowledgements

This work was supported by grant 31003A-182519 from the Swiss National Science Foundation. We particularly thank Nicole Beuret, Janine Bögli from the Biozentrum FACS Core Facility (FCF), Kai D Schleicher and Alexia Ferrand from the Biozentrum Imaging Core Facility (IMCF), for their support, and Scottie Robinson and Jennifer Hirst for the HeLaM-GGA2ks cells and advice on the knocksideways system.

### Author Contributions

DP Buser: conceptualization, resources, data curation, formal analysis, supervision, validation, investigation, methodology, project administration, and writing—original draft, review, and editing.
G Bader: formal analysis and investigation.
M Spiess: conceptualization, resources, data curation, formal analysis, supervision, funding acquisition, validation, visualization, methodology, project administration, and writing—original draft, review, and editing.

### Conflict of Interest Statement

The authors declare that they have no conflict of interest.

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
