## [Reviewer comments · Life Science Alliance]

Life Science Alliance

Retrograde transport of CDMPR depends on several machineries as analyzed by sulfatable nanobodies

Dominik Buser, Gaétan Bader, and Martin Spiess

DOI: <https://doi.org/10.26508/lsa.202101269>

Corresponding author(s): *Dominik Buser, University of Basel*

Review Timeline:

Submission Date:	2021-10-22
Editorial Decision:	2021-10-26
Revision Received:	2022-02-13
Editorial Decision:	2022-03-07
Revision Received:	2022-03-08
Accepted:	2022-03-09

Scientific Editor: Novella Guidi

Transaction Report:

Please note that the manuscript was previously reviewed at another journal and the reports were taken into account in the decision-making process at Life Science Alliance. Since the original reviews are not subject to Life Science Alliance's transparent review process policy, the reports and author response cannot be published.

October 26, 2021

Re: Life Science Alliance manuscript #LSA-2021-01269-T

Dr. Dominik P. P. Buser
University of Basel
Biozentrum
Spitalstrasse 41
Basel 4056
Switzerland

Dear Dr. Buser,

Thank you for submitting your manuscript entitled "Retrograde transport of CDMPR depends on several machineries as analyzed by sulfatable nanobodies" to Life Science Alliance. The manuscript was previously submitted and reviewed at another journal. The authors then chose to transfer their manuscript, along with the reviewers' comments and a proposed revised plan to Life Science Alliance (LSA). The reviewer comments and revision plan was assessed at LSA, and LSA editors deemed that the manuscript could be further considered at LSA provided the authors revise the manuscript, in accordance to what they have laid out in the pbp rebuttal / revision plan. We, thus, encourage you to submit a revised manuscript to us that includes all the experiments you have laid out in your Revision plan.

Thank you for this interesting contribution to Life Science Alliance. We are looking forward to receiving your revised manuscript.

Sincerely,

B. MANUSCRIPT ORGANIZATION AND FORMATTING:

March 7, 2022

RE: Life Science Alliance Manuscript #LSA-2021-01269-TR

Dr. Dominik P. P. Buser
University of Basel
Biozentrum
Spitalstrasse 41
Basel 4056
Switzerland

Dear Dr. Buser,

Thank you for submitting your revised manuscript entitled "Retrograde transport of CDMPR depends on several machineries as analyzed by sulfatable nanobodies". We would be happy to publish your paper in Life Science Alliance pending final revisions necessary to meet our formatting guidelines.

- we encourage you to revise the figure legend for figure S4 such that the figure panels correspond with the figure;
- please add callouts for Figures S2A-B, S3A-D, and S4A-B to your main manuscript text
- Please indicate molecular weight next to each protein blot

A. FINAL FILES:

B. MANUSCRIPT ORGANIZATION AND FORMATTING:

Sincerely,

March 9, 2022

RE: Life Science Alliance Manuscript #LSA-2021-01269-TRR

Dr. Dominik P. Buser
University of Basel
Biozentrum
Spitalstrasse 41
Basel 4056
Switzerland

Dear Dr. Buser,

Thank you for submitting your Research Article entitled "Retrograde transport of CDMPR depends on several machineries as analyzed by sulfatable nanobodies". It is a pleasure to let you know that your manuscript is now accepted for publication in Life Science Alliance. Congratulations on this interesting work.

DISTRIBUTION OF MATERIALS:

Again, congratulations on a very nice paper. I hope you found the review process to be constructive and are pleased with how the manuscript was handled editorially. We look forward to future exciting submissions from your lab.

Sincerely,
